# Epigenomic profiling of primate lymphoblastoid cell lines reveals the evolutionary patterns of epigenetic activities in gene regulatory architectures

Raquel García-Pérez [1,11✉], Paula Esteller-Cucala [1,11], Glòria Mas[2,3], Irene Lobón[1], Valerio Di Carlo [2,3], Meritxell Riera[1], Martin Kuhlwilm [1], Arcadi Navarro [1,4,5], Antoine Blancher [6,7], Luciano Di Croce [2,3,5], José Luis Gómez-Skarmeta[8], David Juan [1,11✉] & Tomàs Marquès-Bonet [1,5,9,10,11✉]

Changes in the epigenetic regulation of gene expression have a central role in evolution. Here, we extensively profiled a panel of human, chimpanzee, gorilla, orangutan, and macaque lymphoblastoid cell lines (LCLs), using ChIP-seq for five histone marks, ATAC-seq and RNA-seq, further complemented with whole genome sequencing (WGS) and whole genome bisulfite sequencing (WGBS). We annotated regulatory elements (RE) and integrated chromatin contact maps to define gene regulatory architectures, creating the largest catalog of RE in primates to date. We report that epigenetic conservation and its correlation with sequence conservation in primates depends on the activity state of the regulatory element. Our gene regulatory architectures reveal the coordination of different types of components and highlight the role of promoters and intragenic enhancers (gE) in the regulation of gene expression. We observe that most regulatory changes occur in weakly active gE. Remarkably, novel human-specific gE with weak activities are enriched in human-specific nucleotide changes. These elements appear in genes with signals of positive selection and human acceleration, tissue-specific expression, and particular functional enrichments, suggesting that the regulatory evolution of these genes may have contributed to human adaptation.

[1] Institute of Evolutionary Biology (UPF-CSIC), PRBB, Barcelona, Spain. [2] Centre for Genomic Regulation (CRG), The Barcelona Institute of Science and Technology, Barcelona, Spain. [3] Universitat Pompeu Fabra (UPF), Barcelona, Spain. [4] National Institute for Bioinformatics (INB), PRBB, Barcelona, Spain. [5] Institució Catalana de Recerca i Estudis Avançats (ICREA), Barcelona, Spain. [6] Laboratoire d'immunologie, CHU de Toulouse, Institut Fédératif de Biologie, hôpital Purpan, Toulouse, France. [7] Centre de Physiopathologie Toulouse-Purpan (CPTP), Université de Toulouse, Centre National de la Recherche Scientifique (CNRS), Institut National de la Santé et de la Recherche Médicale (Inserm), Université Paul Sabatier (UPS), Toulouse, France. [8] Centro Andaluz de Biología del Desarrollo (CABD), Consejo Superior de Investigaciones Científicas-Universidad Pablo de Olavide-Junta de Andalucía, Seville, Spain. [9] CNAG-CRG, Centre for Genomic Regulation (CRG), Barcelona Institute of Science and Technology (BIST), Barcelona, Spain. [10] Institut Català de Paleontologia Miquel Crusafont, Universitat Autònoma de Barcelona, Cerdanyola del Vallès, Barcelona, Spain. [11] These authors contributed equally: Raquel García-Pérez, Paula Esteller-Cucala, David Juan, Tomàs Marquès-Bonet. ✉email: raquel.garcia@bsc.es; david.juan@upf.edu; tomas.marques@upf.edu

Changes in chromatin structure and gene regulation play a crucial role in evolution[1,2]. Gene expression differences have been extensively studied in a variety of species and conditions[3–9]. However, there is still much unknown about how regulatory landscapes evolve, even in closely related species. Previous work has focused on the dynamics of the addition and removal of RE with signals of strong activity during mammalian evolution—mainly defined from ChIP-seq experiments on a few histone marks[10–14]. These analyses suggested that enhancers evolve faster than promoters. The number of active enhancers located near a gene—its regulatory complexity—has also been reported to influence the conservation of gene expression in mammals[10].

Moreover, in a selected group of primates—mostly chimpanzees and macaques—changes in histone mark enrichments are associated with gene expression differences[15–18]. Several studies have also targeted the appearance of human-specific methylation[19–23] patterns and active promoters and enhancers in different anatomical structures and cell types[11–15]. All these studies have proven that comparative epigenomics is a powerful tool to investigate the evolution of RE[24,25]. However, a deeper understanding of the evolution of gene regulation requires the integration of multilayered epigenome data. Only such integration can provide the necessary resolution of regulatory activities for investigating recent evolutionary time frames, as is the case within the primate lineage. Here, we provide an in-depth comparison of the recent evolution of gene regulatory architectures using a homologous cellular model system in human and non-human primates. We observe different levels of correlation between sequence and epigenetic conservation for RE with different activities and highlight the contribution of intragenic enhancers (gE) to explain gene expression levels. We also report the role that often understudied epigenetic states may have in recent human evolution.

## Results

### Comprehensive profiling of primate lymphoblastoid cell lines (LCLs).
We have extensively characterized a panel of lymphoblastoid cell lines (LCLs) from human, chimpanzee, gorilla, orangutan, and macaque, including two independent biological replicates for each species. This characterization includes chromatin immunoprecipitation data (ChIP-seq) from five key histone modifications (H3K4me1, H3K4me3, H3K36me3, H3K27ac, and H3K27me3) and deep-transcriptome sequencing (RNA-seq) (Fig. 1). We integrate these datasets into gene regulatory architectures (Fig. 2a and Supplementary Figs. 1, 2) to (1) understand how primate gene expression levels are controlled and how expression changes between species occur and to (2) study patterns of evolutionary conservation of RE in primates. To complement this resource, we have also processed high coverage whole-genome and whole-genome bisulfite sequencing data, as well as chromatin accessibility data (Supplementary Tables 1–6 and Supplementary Data 1–4). Taken together, this is the most extensive collection of great apes and macaque transcriptomic and epigenomic data to date.

### Annotation of RE.
We used the signal of the ChIP-seq experiments from the five histone marks to identify putative regulatory regions with characteristic marks of promoters or enhancers (Supplementary Figs. 1 and 2). We defined regulatory regions for each cell line as those containing chromatin states (over-represented combinations of histone marks detected by ChromHMM[26]) enriched in any regulatory-related histone mark (Methods and Fig. 2a and Supplementary Fig. 1). We merged

overlapping regulatory regions in the two replicates of every species to define species RE.

We classified the chromatin states of the RE based on a hierarchy of functionally interpretable epigenetic states. This hierarchy differentiates chromatin states into promoter (P) and enhancer (E) states, with three different levels of activity each: strong (s), poised (p), or weak (w) (Methods and Supplementary Fig. 1). We improved these assignments by applying a linear discriminant analysis (LDA) with normalized histone and open chromatin enrichments (Methods and Supplementary Figs. 3 and 4). The refined classification results in more similar regulatory landscapes between biological replicates (Wilcoxon signed-rank test: $P < 0.05$ in all species; Supplementary Fig. 5), increasing the number of RE with the same state in all species (Wilcoxon signed-rank test: $P = 0.03$; Supplementary Fig. 6). At large, promoters have a positive H3K4me3/H3K4me1 ratio, whereas H3K4me1 is more abundant in enhancers than promoters. RE with strong activities have an H3K27ac enrichment level similar to that of H3K4me3 in promoters and H3K4me1 in enhancers. Poised RE have a characteristic enrichment of H3K27me3. Finally, weak promoters and enhancers are associated with lower intensity enrichments of all epigenetic signals and generally lack H3K27me3 and H3K36me3 (Supplementary Fig. 3).

On average, we found ~11,000 and ~76,000 RE with promoter and enhancer states per species, respectively (Fig. 2b), of which 69% and 33% are strong, 8% and 4% are poised, and 14% and 45% are weak, respectively (Supplementary Fig. 7 and Supplementary Data 1). Strong and poised activities are more associated with promoter states, whereas weak activities are more frequently associated with enhancer states (Chi-square test: $P < 2.2 \times 10^{-16}$ in all species). We associated RE with genes using 1D gene proximity and existing high-resolution 3D chromatin contact data for one of the human LCLs (Fig. 2a and Methods). On average, 70% of the RE are associated with genes, of which an average across species of 93% of the RE are associated with protein-coding genes and 61% of the RE are associated with one-to-one orthologous protein-coding genes in all primate species (Fig. 2c). The set of RE associated with a gene defines its regulatory architecture.

Altogether, this catalog of RE provides a comprehensive view of the regulatory landscape of LCLs in humans and nonhuman primates. In contrast to other commonly used definitions of promoters and enhancers limited to strongly active regions, our multilayered integration approach allows the additional annotation of weak and poised activities[11,13]. These activities are of particular relevance to improve the definition of elements in regulatory gene architectures. In sum, a detailed primate regulatory catalog enables the study of the evolution of these regulatory activities using LCLs as a proxy of their regulatory potential in other cell types or conditions.

### The evolutionary dynamics of promoters and enhancers in primate LCLs recapitulate previous observations in more distant mammals.
Interspecies differences in regulatory regions can be associated with genomic or epigenetic changes. Inconsistencies in the quality of genome assemblies make it difficult to distinguish actual interspecies genomic differences, an issue aggravated in multispecies comparisons. To overcome this problem, we restricted our analyses to unambiguous one-to-one orthologs between all species. We detected 28,703 one-to-one orthologous genomic regions in the five species with a promoter or enhancer state in at least one species (Supplementary Fig. 8). Most of these orthologous regulatory regions (~76%, Binomial test: $P < 2.2 \times 10^{-16}$) are associated with genes (Methods). In downstream analyses, we focused on these regions integrating the regulatory architectures of protein-coding and non-coding genes.

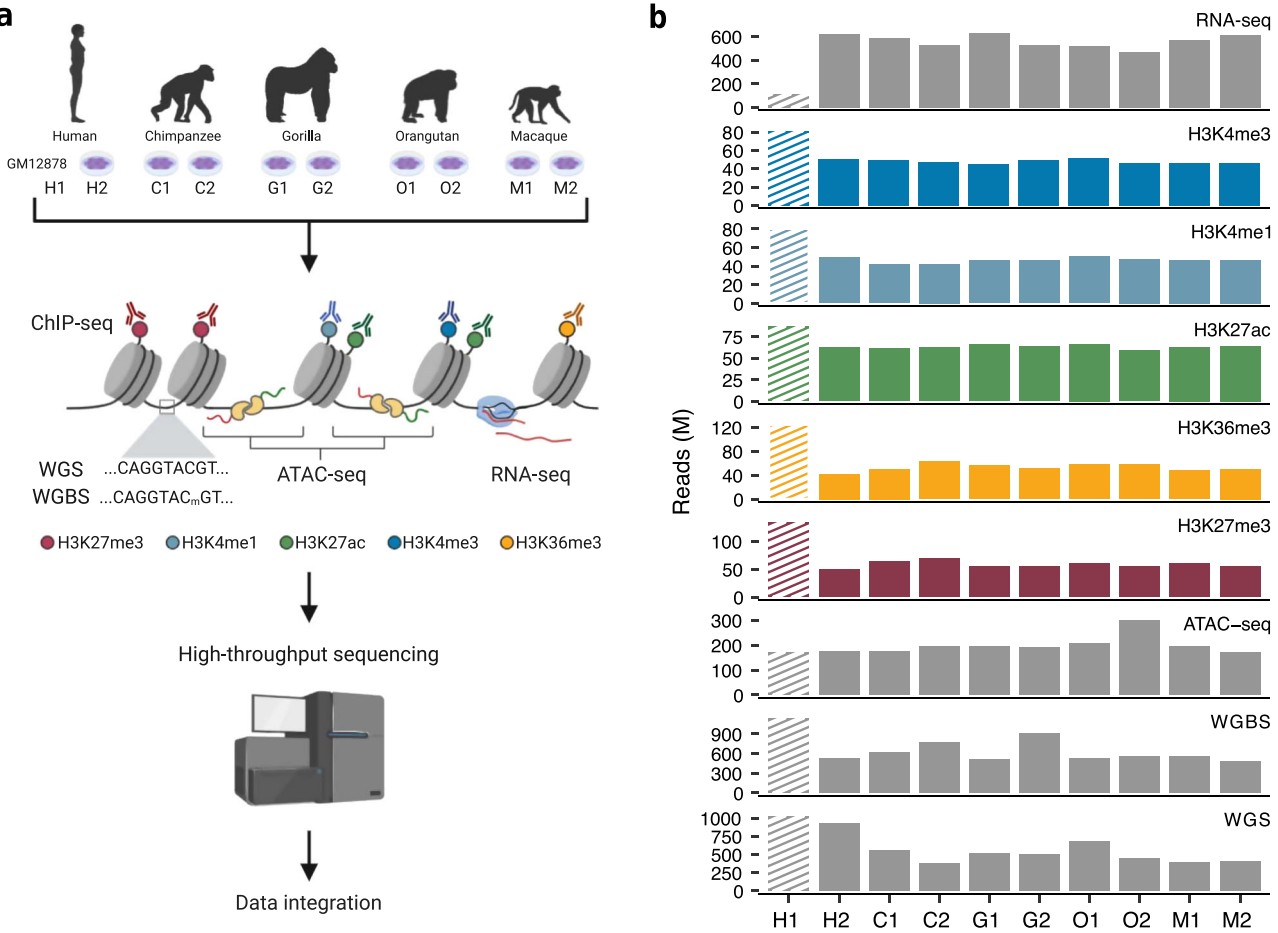

**Fig. 1 Overview of the study design and data generated. a** One human and eight nonhuman primate lymphoblastoid cell lines (LCLs) were cultured to perform a variety of high-throughput techniques, including whole genome sequencing (WGS), whole genome bisulfite sequencing (WGBS), chromatin-accessibility sequencing (ATAC-seq), chromatin immunoprecipitation sequencing (ChIP-seq) targeting five different histone modifications (H3K27me3, H3K4me1, H3K27ac, H3K4me3, and H3K36me3) and transcriptome sequencing (RNA-seq). We integrated previously published datasets from an extensively profiled human LCL (GM12878) to balance the number of human samples. **b** Number of sequencing reads generated per sample and experiment. Striped lines indicate data retrieved from previously published experiments[79,108].

We quantified the conservation of epigenetic states in regulatory regions as the number of primate species with the epigenetic state in the orthologous regions. In the regulatory architectures of protein-coding genes, promoter states are more conserved than enhancer states (Supplementary Fig. 9), with 73% and 60% of regions with a promoter or enhancer state being fully conserved across primates, respectively (two-tailed Fisher's exact test: $P < 2.2 \times 10^{-16}$, OR = 1.48). Less than 14% and 8% of orthologous regulatory regions with a promoter or enhancer state are specific to a primate species. These results for protein-coding genes fall in line with the higher conservation of promoters previously observed in mammals[13]. In contrast, for non-coding genes, promoter states are less conserved than enhancer states (two-tailed Fisher's exact test: $P < 2.2 \times 10^{-16}$, OR = 0.39; Supplementary Fig. 9), with 46% and 69% of fully conserved and 26% and 3% of species-specific elements, respectively.

Intrigued by the different epigenetic conservation patterns in protein-coding and non-coding genes, we studied the repurposing and acquisition of RE. We defined recently repurposed promoters—or enhancers—as regulatory regions with a promoter state in only one species and enhancer states in the remaining species—or vice versa. Similarly, recently gained promoters or enhancers are those regions with a promoter or enhancer state in one species and without regulatory states in any other species.

In agreement with previous studies in more distant species[27,28], nearly all (93%) recently evolved promoter states are acquired through repurposing events, whereas the majority (90%) of recently evolved enhancer states are gained (Chi-square test: $P < 2.2 \times 10^{-16}$; Methods and Supplementary Fig. 10). The regulatory architectures of protein-coding and non-coding genes—the latter evaluated in human due to underrepresentation of non-coding gene annotations in nonhuman species—show this same pattern (two-tailed Fisher's exact test: $P < 2.2 \times 10^{-16}$, OR = Inf, and $P = 6.6 \times 10^{-7}$, OR = 137 respectively; Supplementary Fig. 10).

Our results confirm those found in more distant species[13,27] and reinforce the generality of these evolutionary dynamics in protein-coding genes. The acquisition of regulatory states in the regulatory architectures of non-coding genes resembles that of protein-coding genes. However, the lower conservation of promoter states associated with non-coding genes suggests that their overall higher conservation is not an intrinsic characteristic of promoter states and that it depends on their specific regulatory relevance in different genes.

**The activity of promoters and enhancers influences their epigenetic and sequence conservation.** Taking advantage of our classification of promoters and enhancers into three different activities (strong, poised, and weak), we further explored the

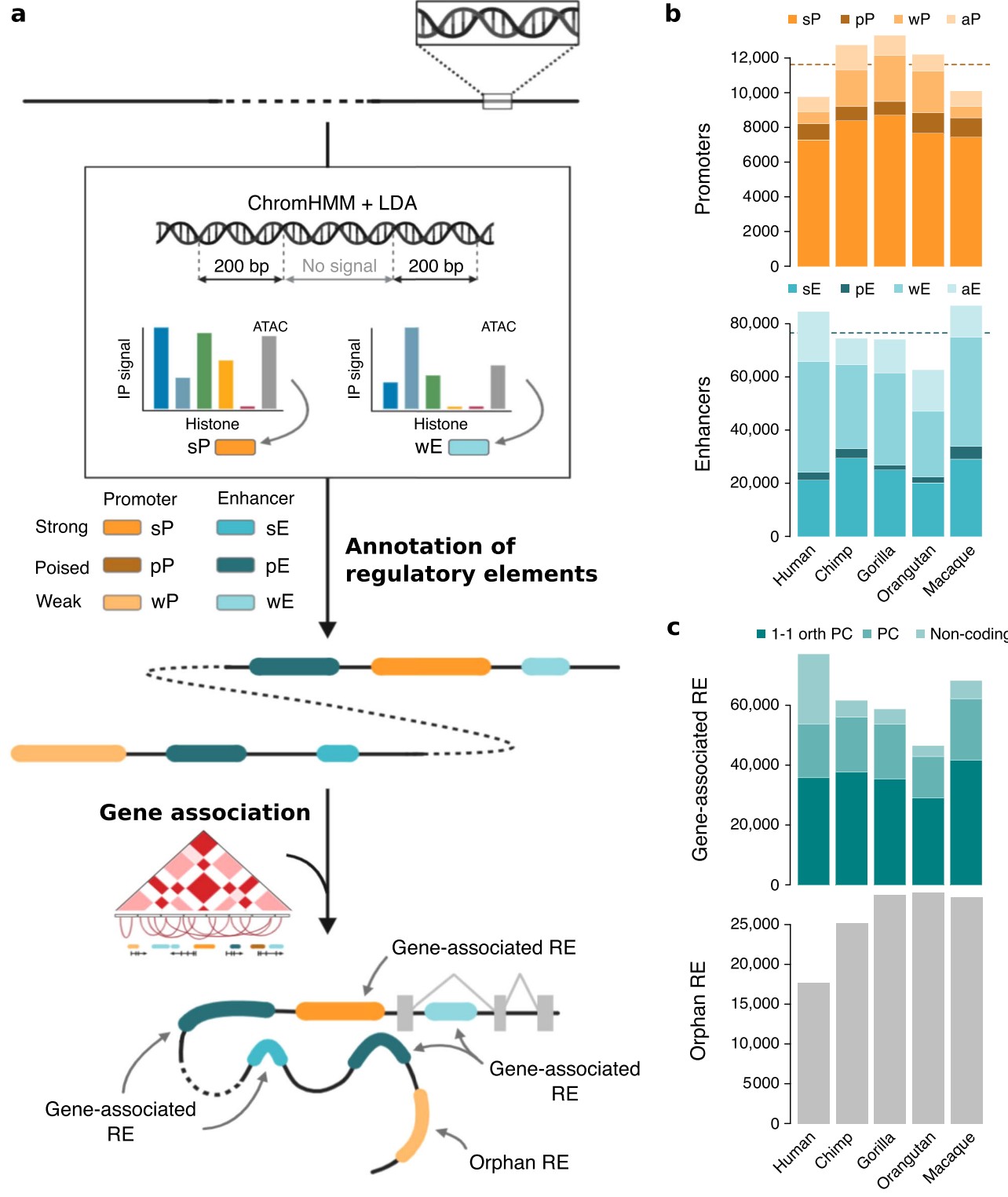

patterns of evolutionary conservation of the different regulatory states. Globally, orthologous regulatory regions conserve their regulatory state (Randomization analyses: 1000 simulations, $P < 0.05$; Supplementary Table 7), but different promoter and enhancer activities show characteristic patterns of conservation (Kruskal–Wallis test: $P < 2.2 \times 10^{-16}$; Fig. 3a and Supplementary Fig. 11).

Strong promoters are the most conserved activities: 80% of them are fully conserved in primates. On the contrary, poised and weak promoters are poorly conserved (Fig. 3a). All enhancer

activities show a similar pattern of evolutionary conservation (Fig. 3a). Enhancer states with strong activities are second in conservation after strong promoters. Nearly 40% of the orthologous regulatory regions with strong enhancer states are fully conserved. Poised enhancers follow closely, with 36% of them conserved in the five species. Lastly, around 21% of the regions with a weak enhancer conserve their activity across primates. The regulatory regions associated with protein-coding and non-coding genes show the same conservation trends (Supplementary Fig. 12). However, strong activities in promoter

**Fig. 2 Epigenetic and regulatory characterization of RE annotated in primates. a** Approach followed to annotate and classify RE. In short, we classify RE in promoter and enhancer states with three activity levels (strong, poised, or weak) based on a combination of chromatin marks and ATAC-seq signals. Bars represent relative enrichment of epigenetic signals in RE, from left to right: H3K4me3 (dark blue), H3K4me (light blue), H3K27ac (green), H3K36me3 (yellow), H3K27me3 (red), and open chromatin (gray). Promoters have a positive H3K4me3/H3K4me1 ratio, whereas H3K4me1 is more abundant in enhancers. RE with strong activities are associated with high H3K27ac levels and poised RE have a robust enrichment in H3K27me3. RE are then linked to genes based on 1D gene proximity and 3D published chromatin maps for LCLs. RE not associated with any gene are referred to as orphan RE. See Methods and extended representation in Supplementary Fig. 1. **b** Number of RE with promoter and enhancer epigenetic states in each species. aP and aE refer to ambiguous promoters and enhancers, respectively. Ambiguous RE are defined as those with a consistent state but different activities between replicates. Dashed lines indicate the average number of RE with promoter and enhancer states annotated across species. **c** Number of RE associated with genes and orphan RE in each species. Genes are divided into one-to-one orthologous protein-coding (1–1 orth PC), protein-coding (PC), and non-protein-coding genes.

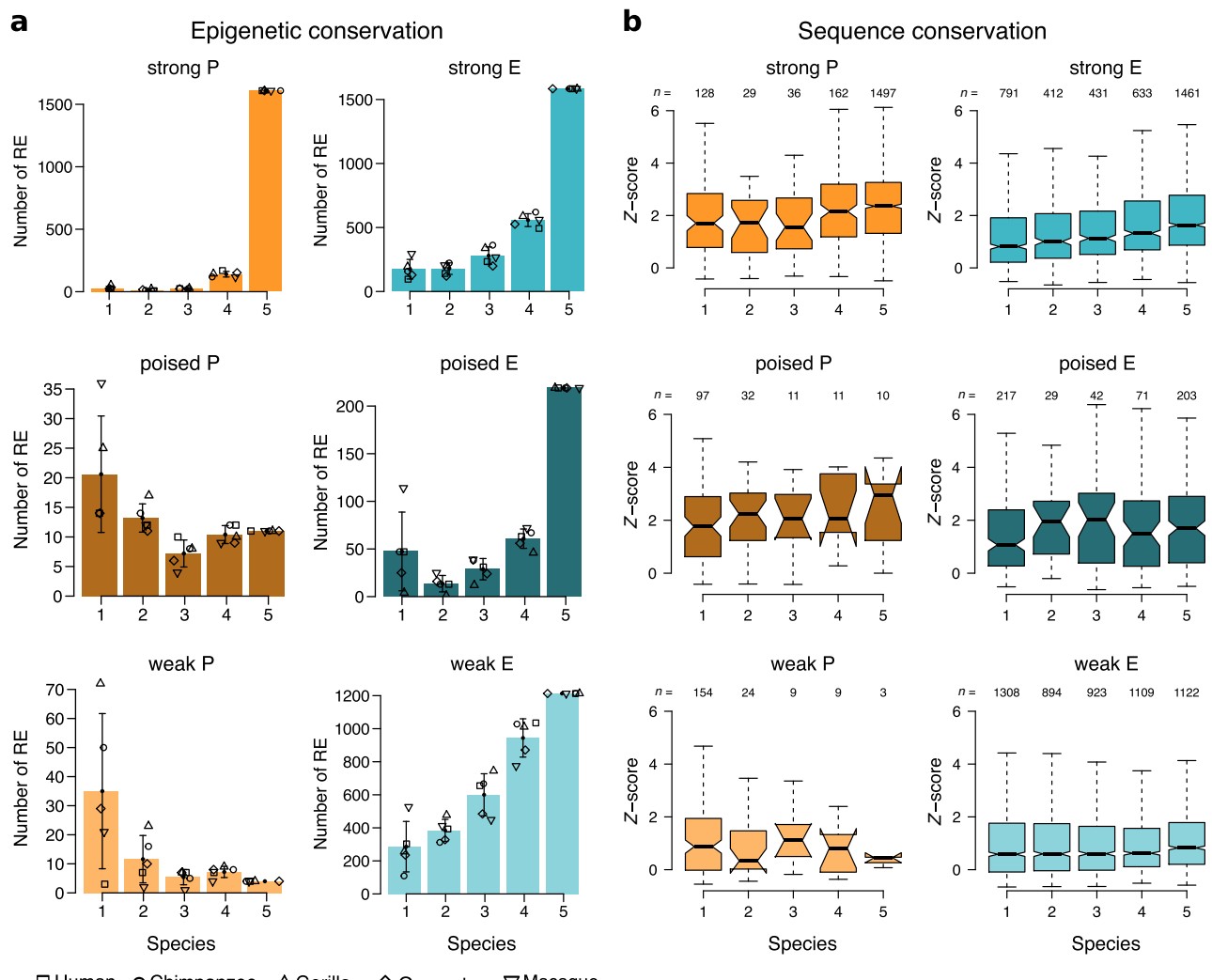

**Fig. 3 Different regulatory activities have different patterns of epigenetic and sequence conservation. a** Bar plots show the average number of orthologous regulatory regions across species with the corresponding color-coded epigenetic state conserved in 1, 2, 3, 4, or 5 species. Points indicate average values and the error bars represent the s.d. (n = 5 species). **b** Distribution of the sequence conservation scores (calculated as Z-scores of the distribution of phastCons30way[29] values for non-coding regions in the same TAD[85]; Methods) of human orthologous regulatory regions with different epigenetic states conserved in 1, 2, 3, 4, or 5 of our primate species. Box plots show medians and the first and third quartiles (the 25th and 75th percentiles), respectively. The upper and lower whiskers extend the largest and smallest value no further than 1.5 × IQR.

states are less common for non-coding than for protein-coding genes, leading to lower conservation of promoter compared to enhancer states. This shows that differences in activity composition can lead to differences in the conservation of the regulatory architectures.

The epigenetic states in a given cell type and their evolutionary conservation reflect the specific function of the regulatory regions

in this cell type. These regions are expected to show different epigenetic states in other cell types, and so their evolutionary patterns might also be different. To investigate whether changes in activity are likely to affect the epigenetic conservation of RE, we assessed the association between epigenetic and sequence conservation—which is cell type-independent. First, we observed that epigenetic conservation significantly correlates with the

conservation of the underlying sequence—quantified as $Z$-scores of background normalized PhastCons values[29]—in all epigenetic states but weak promoter states (Fig. 3b, Methods, and Supplementary Figs. 13 and 14). These correlations are seen in the architectures of protein-coding but not in non-coding genes (Randomization analyses: 1000 simulations; Fig. 3b and Supplementary Fig. 15). Of note, orthologous regulatory regions with fully conserved epigenetic states show significant differences in sequence conservation (Kruskal–Wallis test: $P < 2.2 \times 10^{-16}$; Supplementary Fig. 16). In particular, strong and weak promoters are associated with higher and lower sequence conservation scores, respectively, whereas all enhancer states range in between these values (Dwass–Steel–Critchlow–Fligner test). The sequence conservation scores associated with strong and poised enhancers are not significantly different. Orthologous regions associated with non-coding genes are fewer and less epigenetically conserved (Supplementary Figs. 12 and 14), which could explain the lack of correlation between the conservation of the sequence and the epigenetic state observed in all but strong enhancers (Supplementary Fig. 15).

These results demonstrate that a detailed classification of promoters and enhancers with different activities into regulatory architectures provides a deeper understanding of their evolutionary constraints and dynamics, expanding previous observations in mammals[13] that could mostly be made for active regulatory activities. The consistent association of epigenetic and sequence conservation also suggests that the epigenetic conservation observed in LCLs is a good proxy for the conservation of the regulatory activity of these elements in our primate species.

**Definition of different types of components in the regulatory architectures**. To characterize the evolution of RE based on their specific role in gene expression, we classified RE into five different components according to their role in the gene regulatory architectures (Fig. 4a, Methods). We first classified RE based on their proximity to a gene into three types of components: genic promoters (gP), gE, and proximal enhancers (prE). As gene expression is controlled by a combination of short- and long-distance regulatory interactions[30], we used available 3D chromatin contact maps for human LCLs[31–33] to link interacting RE to their target gene/s and define two additional types of components: promoter-interacting enhancers (PiE) and enhancer-interacting enhancers (EiE) (Fig. 4a).

We were able to link to genes and classify, on average, nearly 3500 otherwise orphan distal RE per species (Supplementary Fig. 17). We annotated ~12,500 gP, ~35,000 gE, ~6700 prE, ~6200 PiE, and ~1800 EiE per species (Fig. 4b and Supplementary Fig. 18), of which 48%, 69%, 40%, 62%, and 61% are associated with one-to-one orthologous protein-coding genes in all primate species (Fig. 4c).

To assess the consistency of our classification of regulatory components, we focused on one-to-one orthologous protein-coding genes considering all their associated RE (i.e., 6 epigenetic states x 5 components = 30 regulatory subcategories). We found high concordance between the epigenetic state (based on ChIP-seq and ATAC-seq data, Fig. 2a) and the component (based on the type of association with the gene, Fig. 4a) of the RE. On average, 75% of gP have a promoter state, and 90% of gene-associated enhancers have an enhancer state (two-tailed Fisher's exact test: $P < 2.2 \times 10^{-16}$ in all species, average OR = 64; Supplementary Fig. 19). This concordance is also consistent across species (Chi-square test: $P < 2.2 \times 10^{-16}$ in all species; Fig. 4d). gP are enriched in RE with strong promoter and poised promoter and enhancer states. Strong enhancers are mostly enriched at gE and PiE, whereas weak enhancers are strongly associated with prE (Supplementary Fig. 19).

Gene expression levels are positively associated with the presence of strong activities in their regulatory architectures and are negatively associated with the presence of poised or weak activities (Kruskal–Wallis test: $P < 0.05$ in all species and regulatory components; Supplementary Fig. 20). These associations are particularly strong in gP and gE (Dwass–Steel–Critchlow–Fligner test; Supplementary Fig. 21). Despite the consistency between the components' activities and gene expression, our results suggest that different types of components might contribute differently to gene expression regulation.

**Regulatory components influence gene expression and its evolution differently**. To explore the ability of our classification of components to discriminate different regulatory roles, we disentangled the underlying network of regulatory co-dependencies between the different regulatory components and gene expression in our cell-type. For this, we used Sparse Partial Correlation Analysis (SPCA)[34] of the normalized RNA-seq and histone mark enrichments (aggregated by promoter and enhancer state in every type of regulatory component) (Methods). This approach establishes a stringent protocol (Benjamini–Hochberg's correction, $P < 1.8 \times 10^{-22}$ for all selected partial correlations) that selects informative partial correlations[34].

To unravel the contribution of each type of component to gene expression, we defined their consensus signal (or eigencomponents) inspired by the notion of eigengenes[35] (Methods). An SPCA based on the eigencomponents shows a consistent global structure of regulatory interactions. gP and gE directly regulate gene expression coordinately, PiE are connected with gPs and EiE with PiE (Fig. 4e and Supplementary Data 5). This regulatory scaffold is consistently observed for the residuals of the histone marks for these eigencomponents (Supplementary Fig. 22 and Supplementary Data 6) when SPCA was performed for all the histone marks together (Supplementary Data 7) and for each of them separately (Supplementary Data 8). To account for the possibility of incompleteness in some of our architectures, we replicated all the analyses using only genes with full regulatory architectures (i.e., genes associated with regulatory components of every type), obtaining consistent results (Supplementary Fig. 23 and Supplementary Data 8).

In agreement with the structure of regulatory interactions recovered by our SPCAs, a generalized linear model of gene expression based on H3K27ac, H3K27me3, and H3K36me3 signals at gP and gE and their interactions (15 variables) explains ~67% of gene expression variability (Supplementary Table 8). Remarkably, this is only 6% lower than an exhaustive naive model, including the signal from all histone marks at all types of regulatory components with all possible interactions (1225 variables) (Supplementary Data 9). These results suggest the epigenetic activities of putatively strong and poised gP and gE and their interactions likely have a large influence on gene expression regulation in our regulatory architectures. However, their co-dependency with the other components suggests that they are dependent, in turn, on the coordination of the whole architecture. Although we cannot infer causality from our SPCA analysis, these networks reflect that regulatory co-dependencies between components depend on the distance of the elements in the network of chromatin contacts (with gP and gE being in the gene locus, PiE interacting directly, and EiE interacting indirectly with it). The robustness of these networks of direct co-dependencies, their ability to explain gene expression, and their correspondence with the spatial disposition of the elements show that these components reflect specific regulatory roles.

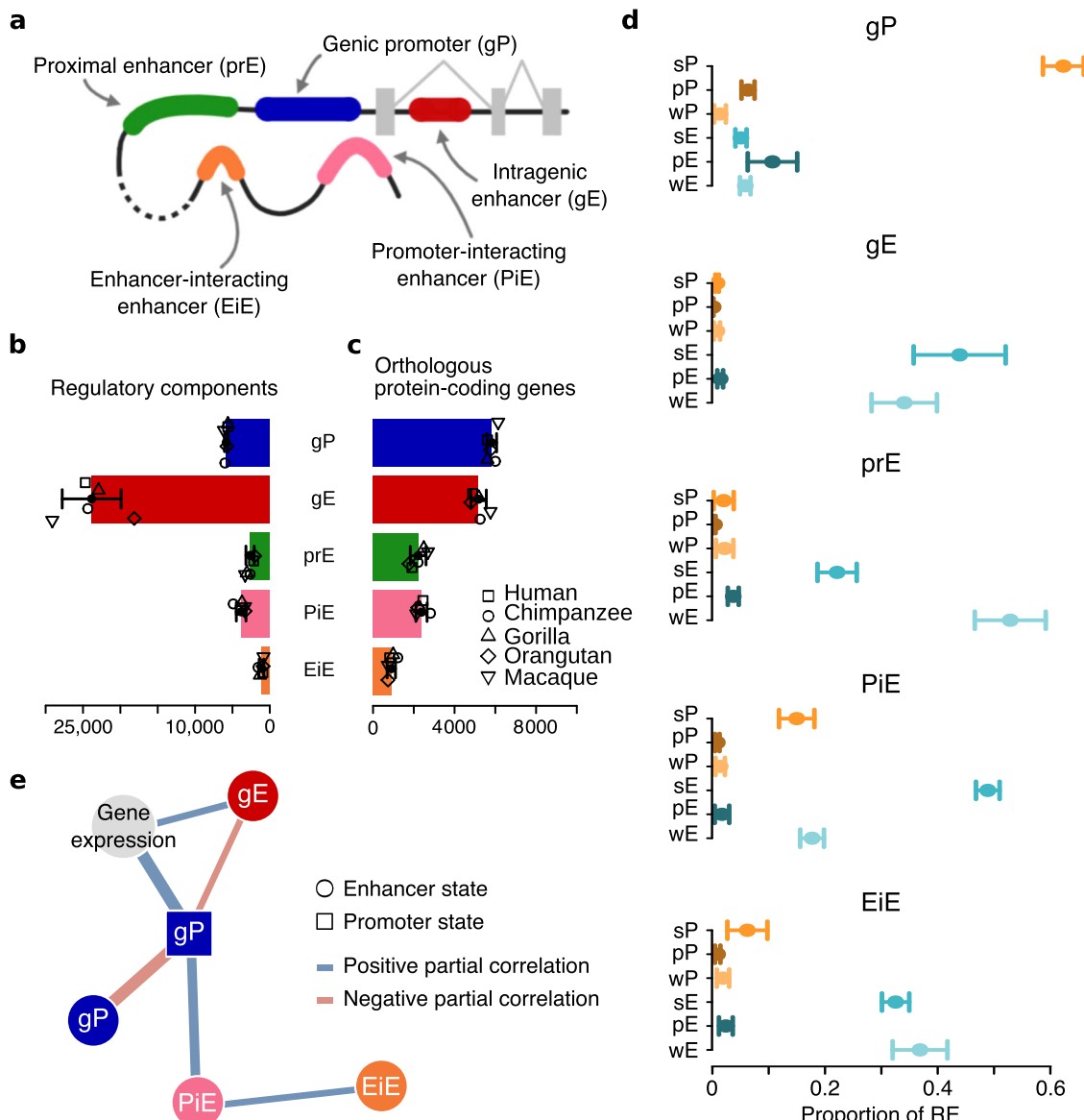

**Fig. 4 Epigenetic signals in gene regulatory architectures explain gene expression levels. a** Classification of RE according to their regulatory roles in gene architectures. Of note, EiE may interact with any type of enhancer in a regulatory architecture (prE, gE, PiE, and EiE). **b** Average number of RE across species associated with one-to-one orthologous protein-coding genes classified as gP, gE, prE, PiE, and EiE. Differently shaped dots show the number corresponding to each species. Points indicate average values and the error bars represent the s.d. ($n = 5$ species). **c** Average number of orthologous protein-coding genes associated with each type of regulatory element. Differently shaped dots show the number corresponding to each species. Points indicate average values and the error bars represent the s.d. ($n = 5$ species). **d** Proportion of RE with a given epigenetic state associated with one-to-one orthologous protein-coding genes for each type of regulatory component. Dots and error bars show the average proportion and s.d. across species ($n = 5$ species), respectively. **e** Sparse Partial Correlation Networks showing the statistical co-dependence of the RNA-seq (Gene expression) and the consensus ChIP-seq signals for the five histone marks in every component represented by the eigencomponents (minimal partial correlation $= -0.41$; maximal partial correlation $= 0.33$; all partial correlations Benjamini–Hochberg's $P < 4.1 \times 10^{-303}$). Edge widths are proportional to absolute partial correlation values within each network. The networks are based on the 5737 one-to-one orthologous protein-coding genes associated with at least one regulatory element in all species. Only nodes for values with significant and relevant partial correlations are represented.

Previous studies have found that gene expression evolution is associated with changes in the regulatory complexity of a gene (the number of nearby RE)[10]. Since we could classify the RE of a gene into different components (Supplementary Fig. 24), we were able to investigate the association of gene expression changes (Supplementary Fig. 25) with the evolutionary differences in the complexity of each type of component. We found that the effect of changes in complexity on gene expression levels depends on the epigenetic state gained or lost and the type of regulatory component affected (Supplementary Fig. 25). Evolutionary

changes that alter the epigenetic state at gP, specifically the presence of either a strong promoter or poised enhancer, as well as the number of gE with either strong or poised enhancer states, show the most robust associations with gene expression differences. The number of prE in any enhancer epigenetic state and strong promoters and strong and poised enhancers in PiE also show significant though modest effects. These results highlight that the additive nature of gene regulation depends on regulatory architectures. This dependency can be captured either by the aggregation of histone enrichment signals (as in our

SPCAs) or by quantifying the number of regulatory components with specific activities. Moreover, they confirm that our regulatory components represent different regulatory roles with a different contribution to gene expression evolution and which evolutionary relevance should be investigated separately.

**Different regulatory components with poised and weak enhancer states appear in LCL-unrelated genes and functions.** We next explored the functional implications that both conserved and species-specific regulatory states in certain components could have (overrepresented combinations in Supplementary Fig. 26). For this, we examined their functional role and tissue-specific expression (GTEx data[36], Supplementary Table 9). We find significant enrichments for the genes targeted by conserved strong promoter states in gP, conserved strong and weak enhancer states in gE, and conserved poised enhancer states in gP and prE (one-tailed Fisher's exact test: Benjamini–Hochberg's correction, false discovery rate (FDR) < 0.05; Methods, Fig. 5a, Supplementary Fig. 27 and Supplementary Data 10). Remarkably, among the genes associated with any species-specific epigenetic state, only those linked to human-specific weak enhancers in intragenic enhancers (hereafter referred to as hswEgE) have significant functional enrichments.

These enrichments show the expected association of conserved strong epigenetic states (strong promoter states in gP and strong enhancer states in gE) with genes involved in relevant cellular processes, such as metabolism, chromatin organization, and regulation of the cell cycle (Fig. 5a, Methods, Supplementary Fig. 27, and Supplementary Data 11 and 12). Functions specific to LCLs like those involving viral processes are specifically enriched in strong enhancers[30]. Moreover and regardless of whether there is a functional enrichment, the components with strong epigenetic states show similar expression profiles across tissues (Fig. 5b and Supplementary Figs. 28–30), with high expression levels in LCLs and most other tissues and wide expression breadth (Fig. 5c and Supplementary Fig. 31).

In contrast, conserved poised enhancer states in gP and prE target genes enriched in developmental and proliferative functions, echoing their known implication in these processes[37–40] (Fig. 5a, Supplementary Fig. 27, and Supplementary Data 13 and 14). Surprisingly, genes with conserved poised enhancer states in their gP are also enriched in neuronal functions and have higher expression levels in brain (Fig. 5b, c and Supplementary Figs. 27–29). Genes associated with both types of conserved poised enhancer states show overall minimal expression levels but high tissue-specificity (median tissue specificity index ($\tau$, Tau) > 0.85 in both; Methods and Supplementary Fig. 31).

Protein-coding genes targeted by gE with conserved weak enhancer states are enriched in various functional annotations, including neuronal ones, such as cell projection and synapse (Fig. 5a, Supplementary Fig. 27 and Supplementary Data 15). This gene set shows low expression in LCLs and high expression in the brain, which is in agreement with its observed functional annotation (Fig. 5b, c). Also, the tissue-specificity of this group is higher than that of conserved strong regulatory activities both in promoters and enhancers (median $\tau = 0.72$, Dwass–Steel–Critchlow–Fligner test: $P < 2.2 \times 10^{-16}$ in the three tests; Supplementary Fig. 31). This apparent brain-specificity is not found in genes associated with conserved weak enhancer states in other regulatory components, which have overall higher expression levels and not particular tissue-specificity, as would be expected from weak epigenetic states[41] (Supplementary Figs. 30 and 31).

Finally, we focused on genes targeted by hswEgE. These genes are solely enriched in neuron parts and synapse (Fig. 5a

and Supplementary Data 16). Similar to genes associated with their analogous conserved group (conserved weak enhancers states in gE), these genes are typically expressed at low levels with their highest expression in tissues unrelated to LCLs, particularly brain, tibial nerve, and testis, while having marginal or no expression in numerous other tissues, including LCLs (Wilcoxon–Nemenyi–McDonald–Thompson test: $P < 1 \times 10^{-4}$; Rank-biserial correlation effect size between brain and LCLs = 0.633; Fig. 5b, c, Supplementary Figs. 28 and 29, and Supplementary Data 17). Remarkably, these genes have higher tissue-specific expression than those with conserved strong activities in their components (median $\tau = 0.84$, Dwass–Steel–Critchlow–Fligner test: $P < 4.5 \times 10^{-14}$; Supplementary Fig. 31).

Intrigued by the high tissue-specificity of the genes with hswEgE, we sought to identify the tissues driving this tissue-specificity taking its analogous conserved group as reference. Testis and brain are the tissues with the highest number of tissue-specific genes ($\tau_{Tissue} > 0.8$), but most interestingly, whereas the fraction of testis-specific genes is comparable between gene sets (two-tailed Fisher's exact test: $P > 0.05$, OR = 1.20), brain-specific genes are more than twofold enriched in genes with human-specific gE (two-tailed Fisher's exact test: $P = 0.02$, OR = 2.29; Supplementary Fig. 31).

Our results show that conserved strong epigenetic states are involved in the regulation of important genes highly expressed in LCLs and other tissues. However, poised and weak enhancer states, either conserved or human-specific, are involved in the regulation of genes marginally expressed in LCLs, but with particular functional roles and tissue-specific expression patterns. These unexpected associations, along with the evolutionary conservation patterns of poised and weak enhancer states, suggest that these epigenetic states might be indicative of putative RE in other cell types different from LCLs.

**Genes with novel human-specific intragenic weak enhancers are targeted by signals of adaptation and nucleotide changes.** The unexpected association of the genes targeted by hswEgE with neuronal functions prompted us to study the relationship these genes might have with signals of positive selection or accelerated evolution in the human genome. In fact, among the genes associated with hswEgE (153 hswEgE in 134 genes, Supplementary Data 18), we found several genes previously proposed to have been subjected to different evolutionary forces. Some of these genes are *FOXP2*, *PALMD*, and *ROBO1*, which have known brain-related functions[42–45] or *ADAM18*[46], *CFTR*[47,48], and *TBX15*[49].

To assess whether genes with hswEgE have been particularly targeted by evolutionary forces, we investigated their co-occurrence in genes associated with signals of positive selection[50–54] and acceleration[55–57], hereafter referred to as putatively selected or accelerated regions (Methods). We find that 41% of the genes with hswEgE (56 genes) are also showing such signatures (one-tailed Fisher's exact test: $P = 3.33 \times 10^{-11}$, OR = 3.5). The results of our nested analysis (Fig. 6a) show a significant association of genes targeted by gE (but not in any other component type), genes targeted by gE with weak enhancer states (but not strong or poised enhancer states), and genes targeted by gE with human-specific weak enhancer states (but not with fully conserved or any other epigenetic states in nonhuman primate species). In addition, no enrichment is observed for genes with gP with conserved poised enhancer states (one-tailed Fisher's exact test: $P > 0.05$, OR = 1.1), even though they also seem to be associated with brain-specific and neuronal functions (Fig. 5).

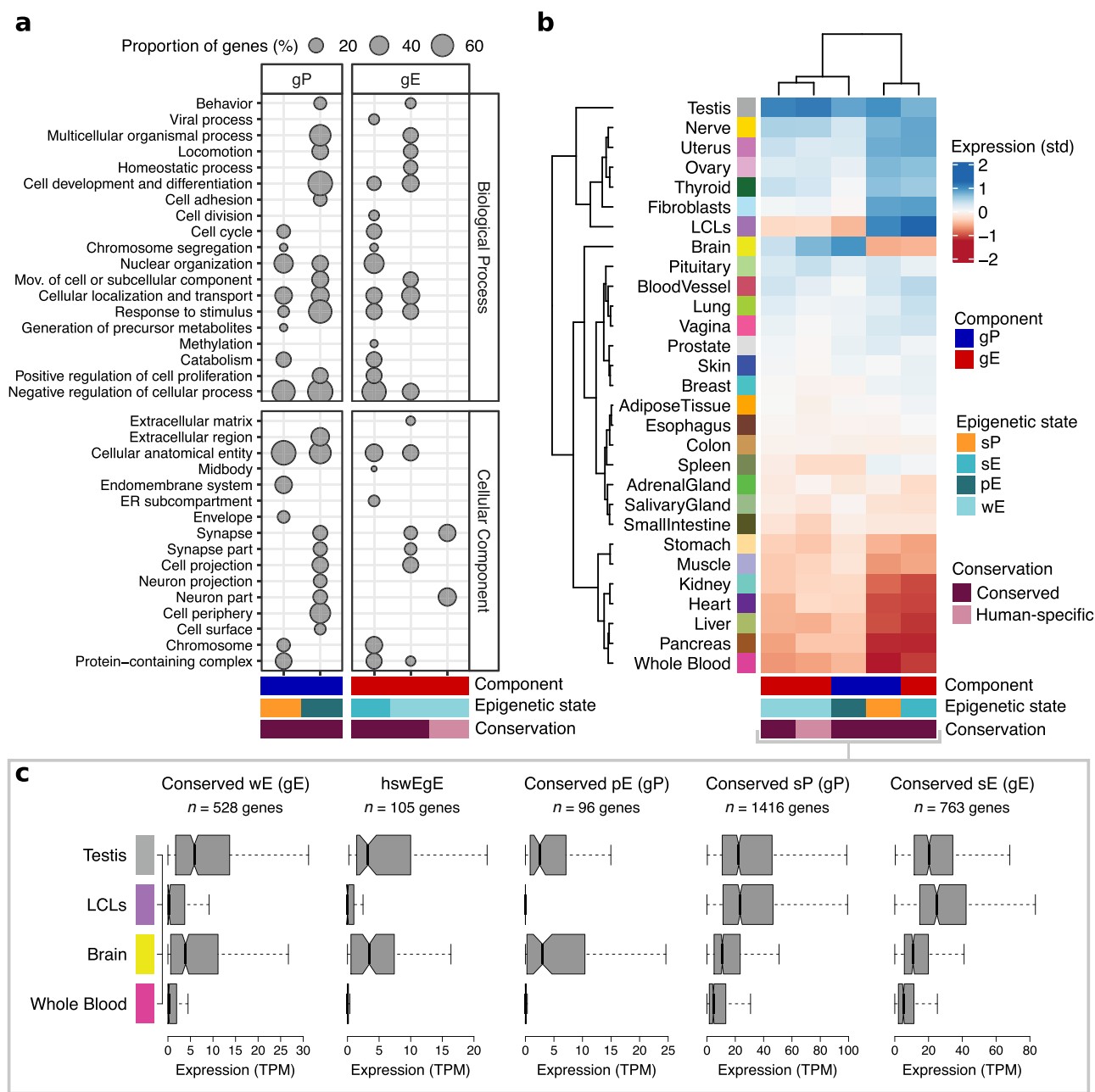

**Fig. 5 Functional enrichment and gene expression patterns indicate that weak and poised enhancer states might be associated with LCL-unrelated cell types. a** Functional enrichment of conserved and human-specific activities in genic promoters and gE. The size of circles indicates the proportion of genes included in each functional category from the total number of genes contained in the corresponding regulatory group. Extended version in Supplementary Fig. 27. **b** Heatmap of standardized expression across tissues in state/component regulatory groups with functional enrichments. Extended version in Supplementary Fig. 28. **c** Median expression levels in testes, LCLs, brain, and whole blood of genes groups in **b**. Box plots show medians and the first and third quartiles (the 25th and 75th percentiles), respectively. The upper and lower whiskers extend the largest and smallest value no further than 1.5 × IQR.

Finally, we explored whether hswEgE could also be associated with human-specific sequence changes. For this, we collected a set of over 2.8 million single nucleotide changes fixed in humans (hSNCs) that differ from fixed variants in the genomes of the remaining nonhuman primates (Methods). Around 22% of the hswEgE-containing genes (30 genes) harbor at least one hSNC in their hswEgE. Also, the density of these human-specific changes is higher in hswEgE compared to gE with conserved weak enhancer states (Mann–Whitney $U$-test: $P = 0.01$; Methods and Supplementary Fig. 32). This observation is not driven by GC-biased gene conversion (gBGC), as the derived nucleotides in

humans are not enriched in W to S mutations compared to the ancestral state (one-tailed Fisher's exact test: $P > 0.05$, OR = 0.9).

Taken together, roughly 10% of the hswEgE-containing genes are in putatively selected or accelerated regions and also harbor hSNCs in these enhancers (14 genes) and the co-appearance of both features is not significantly enriched (one-tailed Fisher's exact test: $P > 0.05$, OR = 1.3, Fig. 6b). This observation indicates that even though putatively selected or accelerated regions and human-specific mutations are both associated with hswEgE, they are not mutually conditioned. As such, it implies that none of these signals alone is required to explain the appearance of hswEgE.

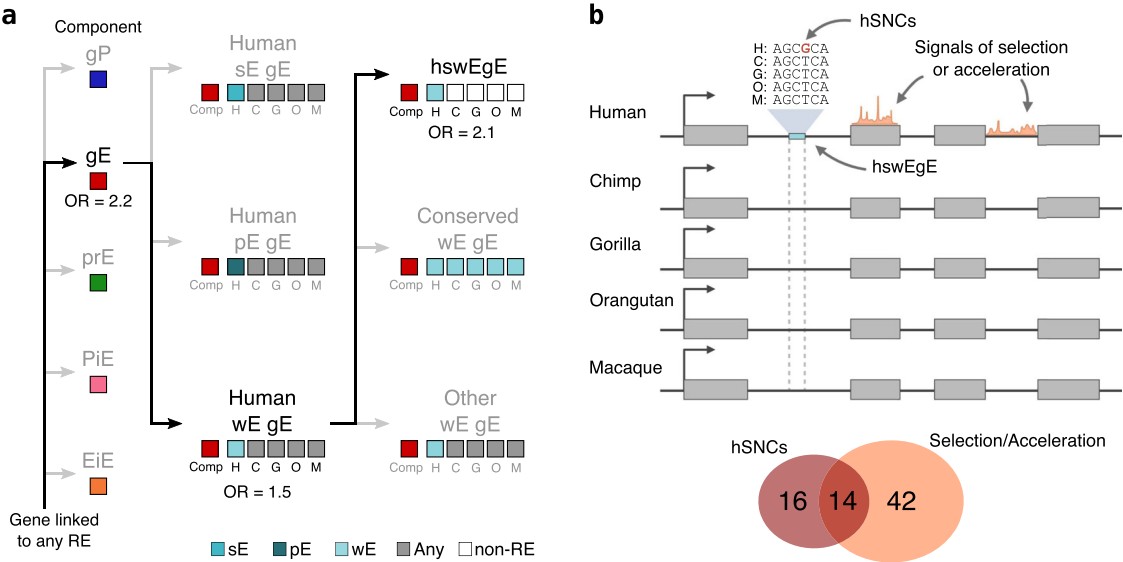

**Fig. 6 Human-specific intragenic enhancers with weak activities (hswEgE) co-localize with signals of recent human selection and acceleration and human-specific nucleotide changes. a** Diagram with the results of the nested enrichment analysis. hswEgE-containing genes are enriched in genes targeted by putative signals of positive selection and acceleration. Epigenetic states for each species and consensus regulatory components are depicted as boxes with the same color code as Figs. 2b and 4a, respectively. **b** Top: Schematic representation of a hswEgE with an hSNC (nucleotide change in humans shown in red) in a gene with signals of selection and acceleration. Bottom: Venn diagram illustrating the overlap between the 56 genes containing hswEgE and with signals of selection or acceleration and the 30 genes with hswEgE and hSNCs.

Among the 14 genes with both human-evolving signals and hSNCs (Fig. 6b), there are several interesting candidates for adaptive evolution associated with different traits. Many of these genes are associated with neuronal functions (*ROBO1, CLVS1, SEMA5A, KCNH7, SDK1*, and *ADGRL2*), but also with pigmentation (*LRMDA*) or actin organization in cardiomyocytes (*FHOD3*). Other interesting genes include instances of hswEgE-containing genes that also are targeted by putatively selected or accelerated regions (*FOXP2, ASTN2, NPAS3*, or *NTM*) or carry hSNCs in these enhancers (*PALMD, VPS13C, IGSF21*, or *CADM2*). Interestingly, we found only one antisense RNA gene, *MEF2C-AS1*, showing both. This gene has been associated with ADHD[58], and its target gene *MEF2C* is a very well-known target of genetic alterations (many of them also affecting *MEF2C-AS1*) associated with severe intellectual disability[59], cerebral malformation[59], or depression[59,60].

Remarkably, three hswEgE accumulate more hSNCs than expected (Randomization test: 10,000 simulations, Bonferroni correction, $P < 0.02$ in all cases; Methods and Supplementary Fig. 32), a number of hswEgE which is also significantly higher than expected (Randomization test: 10,000 simulations, $P = 8 \times 10^{-4}$; Supplementary Fig. 32). Two of these genes are protein-coding genes with known functions in brain cell types and with signals of human adaptation. *CLVS1* is a protein-coding gene with brain-specific expression ($\tau_{\text{Brain}} = 0.964$) required for the normal morphology of endosomes and lysosomes in neurons[61]. *ROBO1* is a broadly expressed integral membrane protein that participates in axon guidance and neuronal migration ($\tau = 0.388$)[62,63] that has also been associated with human speech and language acquisition since the split from chimpanzees[43]. The third enhancer is included in *AC005906.2*, a long intergenic non-protein-coding gene specifically expressed in brain ($\tau_{\text{Brain}} = 1$). Interestingly, this gene overlaps with *KCNA1*, a voltage-gated potassium channel with the same brain-specific expression pattern ($\tau_{\text{Brain}} = 0.995$) and for which mutations have been associated with neurological malfunctions[64].

We show that hswEgE target neuron-related brain-specific genes that are associated with putative signals of positive selection and acceleration and an excess of hSNCs. These human-specific

nucleotide changes are especially concentrated in three of them. Two of these genes, *CLVS1* and *ROBO1*, exemplify the confluence of signals of human adaptation and hSNCs in protein-coding genes that are important for normal neuronal structure, migration, and axon guidance in the human brain.

## Discussion

The evolution of human and nonhuman primates is an area of major interest, but ethical, legal, and practical constraints often limit access to direct biological material. In this study, we have generated a comprehensive and unified dataset of epigenomic landscapes in LCLs for human and four nonhuman primate species. Despite the artificial nature of our cellular model[65–67], previous studies have shown the value of LCLs as an experimentally convenient model of somatic cells that accurately resembles the phenotype of its cell type of origin[68] and which can be robustly used for comparative studies in humans and primates[18,69–71]. Moreover, its clonality ensures a cell type-specific experimental system reducing the confounding factors associated with cell population diversity in bulk tissue samples.

Using this cell model, we reproduced previous observations on the dynamics of the evolution of RE reported in more distant species[10,13,27], which we show can be extrapolated to closely related species (at least for great apes and macaques). Moreover, we have expanded these observations to explain how these dynamics result from the different evolutionary constraints associated with their epigenetic activities. Therefore, we show that considering weak and poised activities is of major relevance to fully understand the evolution of regulatory regions.

We also observed that different epigenetic activities have characteristic evolutionary patterns with higher conservation for strong promoter and strong and poised enhancer states. The correlations between epigenetic and sequence conservations are also different for each epigenetic state, with higher correlations for strong and poised promoter and enhancer states. These differences are likely due to their different influence on gene expression. Therefore, the previously reported higher conservation of promoters probably reflects its often greater influence on

gene expression. We are also able to confirm this hypothesis with the lower conservation of promoter states in the regulatory architectures of the non-coding genes where strong promoter states are scarce.

We further classified the RE in five different components of gene regulatory architectures. These components are genic promoters (gP) and intragenic, proximal, promoter-interacting, and enhancer-interacting enhancers (gE, prE, PiE and EiE, respectively). By studying their regulatory co-dependencies, we observe that the epigenetic activity of each type of component influences gene expression differently. In brief, coordinated epigenetic activities in gP and gE form the core of these architectures and are strongly correlated with gene expression levels. Regulatory activities in PiE are also coordinated with promoter components, and activities in EiE are associated with PiE. With these results, we show that the impact of regulatory components on gene expression reflects the structure of the regulatory architecture.

We find that gene expression changes are associated with the regulatory complexity of gene architectures and with the epigenetic activity of the different regulatory components. Namely, we report that gene expression changes between primate species occur through the addition or removal of strong promoter activities in promoter components or strong and poised enhancer activities in gE. The remaining components show fewer changes directly linked to expression differences, but they can still be instrumental for gene expression evolution, probably through their influence on gP and gE. The gene architectures we present provide a starting point for future in-depth investigations on the interdependence of different regulatory regions and mechanisms in the evolution of gene regulation. In this sense, we stress the importance of considering promoter and enhancer activity states in the different types of gene components to achieve a more detailed description of the regulatory processes.

Despite the large influence of strong activities on gene regulation, our results in LCLs suggest that unforeseen information can be drawn from the analysis of elements with a repressive or negligible regulatory role in our cell model. Functional enrichment analyses show that genes with conserved poised activities are enriched in cell proliferation and differentiation, as shown by others[37–40]. In the case of poised enhancer states in gP, these genes are specifically expressed in brain. Moreover, we find that recently evolved weak gE in the human lineage (hswEgE) occur in genes showing patterns of brain-specific gene expression and enriched in neuronal functions. This tissue-specific activity is consistent with what has been recently reported for gE[72]. These genes are also associated with signals of positive selection and acceleration, suggesting they may have contributed to human adaptation in several traits. However, we acknowledge that these evolutionary signals could also be explained by relaxed selection[73]. These evolutionary patterns are unique to hswEgE and not found in conserved poised enhancers in gP even though these elements also target genes with brain-related functions. Moreover, we show that hswEgE accumulate more human-specific sequence changes than expected. These changes could potentially alter the activity of the enhancer, as has been recently shown[74]. We highlight two interesting protein-coding genes that show the convergence of both signals and an excess of human-specific mutations which have key functional roles in neuronal structure, migration, or axon guidance: CLVS1 and ROBO1.

Using LCLs as our model system, we have seen that poised and weak enhancer activities may carry information about gene regulation in unrelated cell types. Notably, we find that genes in putatively selected or accelerated regions also harbor human-specific epigenetic and sequence innovations. Given the tissue-specificity and functional enrichments of these genes, which are

marginally expressed and unrelated to LCLs, we hypothesize that functionally relevant regulatory innovations that appear in a given cell type could be echoed as weaker activity signals in a different cell type. This could explain our observations in LCLs. As such, our results suggest that human-specific weak enhancer activities could provide an unexpected window for the study of regulatory evolution in the human lineage. Further research will be needed to clarify the specific role of these elements in different tissues and cell types.

Taken together, this work shows that the evolution of gene regulation is deeply influenced by the coordination of epigenetic activities in gene regulatory architectures. Our insights call for the incorporation of better integrative datasets and refined definitions of regulatory architectures in comparative evolutionary studies to fully understand the interplay between epigenetic regulation and gene expression.

## Methods

**Cell line acquisition and cell growth.** Lymphoblastoid cell line (LCL) GM19150 (Yoruban male) was purchased from the Coriell Institute. Chimpanzee, gorilla, and orangutan LCLs[75] were kindly provided by Dr. Antoine Blancher. Macaque LCLs[76,77] were kindly provided by Dr. Gaby Dioxiadis.

Cell lines were grown in suspension at a confluency of 200,000 cells/mL to $1 \times 10^6$ cells/mL, at 37 °C and 5% $CO_2$, in RPMI 1640 media (Invitrogen 42401-018) with 15% fetal bovine serum (FBS Invitrogen 10270-106), 1% penicillin-streptomycin (Invitrogen 15140-122), and supplemented with 2 mM L-glutamine (Sigma G7513-100ML).

**DNA and RNA extractions.** For DNA and RNA extractions, ~5 million cells were pelleted and washed with PBS (Sigma D8537).

DNA extractions were performed using a phenol-chloroform protocol. Briefly, cell pellets were resuspended in a lysis buffer (10 mM Tris-HCl pH 8, Life Technologies 15568-025; 10 mM NaCl, Sigma 71386; 0.2% IGEPAL CA-630, Sigma I8896) and incubated in ice for 30 min. Next, cells were washed and incubated at 37 °C for 30 min in NEB2 buffer (New England Biolabs B7002S), supplemented with 10% SDS (Life Technologies AM9820) and 5 µL of RNAse A (QIAGEN 79254), followed by an overnight incubation at 65 °C with 10 µL Proteinase K (Life Technologies AM2546). Two successive phenol-chloroform extractions were performed the day after. DNA was precipitated with NaAc 3 M and washed twice with 70 and 100% EtOH, respectively. DNA integrity was checked in an agarose gel. Extracted DNA was used for WGS and WGBS libraries. Libraries were prepared using the TruSeq DNA PCR Free Library Preparation Kit and TruSeq DNA Methylation Kit, respectively, following Illumina's standard protocol. One-fifty base pair paired-end (150PE) reads were sequenced in a HiSeqX machine.

RNA was extracted using the miRNeasy Mini kit (QIAGEN 217004) following the manufacturer's instructions. The TruSeq Stranded Total LT Samples Prep Kit with Ribo-Zero was used for library construction, following Illumina's standard protocol. Library preparation and sequencing were performed at Macrogen (Seoul, Korea). Three technical replicates (R1, R2, and R3) were sequenced per sample. All nine libraries per technical replicate were pooled and loaded into one lane. About 101 bp paired-end reads (101PE) were sequenced in a HiSeq4000 machine.

**Chromatin immunoprecipitation.** For ChIP-seq experiments, LCL primate cell lines were harvested and washed with PBS (Sigma D8537). Cells were fixed by incubating in 1.5% formaldehyde (Sigma F8775) PBS buffer at room temperature for 10 min. Formaldehyde was quenched with 125 mM glycine (Invitrogen 15527-013) at room temperature for 5 min, after which the samples were placed on ice for 15 min. Cells were then washed twice in cold PBS, aliquoted, and pellets of ~20 million cells were frozen at −80 °C. ChIP experiments were carried out following a standard protocol in the lab[78]. Briefly, crosslinked cell pellets were resuspended in 500 µL ice-cold ChIP buffer (50 mM Tris-HCl pH 8, 100 mM NaCl, 5 mM EDTA, 0.33% SDS, 1.67% Triton X-100) with Protease inhibitors (cOmplete™ EDTA-free, SIGMA) and 1 mM PMSF. Lysates were sonicated in a Bioruptor (Diagenode) to obtain an average fragment size of 250 bp and centrifuged for 20 min at full speed. An aliquot of the soluble chromatin was taken, reverse cross-linked, purified using the Qiagen PCR purification kit (Cat No./ID: 28104), and quantified by Nanodrop. For each ChIP, 30 µg of soluble chromatin, 0.75 µg of mouse E14TG2a (as spike-in control), and 5 µg of the corresponding histone modification antibody were incubated overnight at 4 °C. ChIP DNA was recovered by incubating with 42 µL protein A bead slurry (Diagenode) for 2 h, washed three times with low salt buffer (50 mM HEPES pH 7.5, 140 mM NaCl, 1% Triton X-100) and one time with high salt buffer (50 mM HEPES pH 7.5, 500 mM NaCl, 1% Triton X-100). DNA complexes were decross-linked at 65 °C overnight with proteinase K (Invitrogen) and DNA was purified using the PCR purification kit (Qiagen, 28104) and quantified by Nanodrop. We used commercially available antibodies against H3K4me3 (Diagenode, C15310003), H3K4me1 (Abcam, ab8895), H3K36me3

(Abcam, ab9050), H3K27me3 (Millipore, 07-449), and H3K27ac (Millipore 07-360).

A total of 63 ChIP-seq libraries were constructed: 45 libraries for the ChIP-seq experiments (one per sample and histone modification) and 18 libraries for Input samples (Input1 for histones H3K4me3, H3K4me1, H3K36me3, and H3K27me3; Input2 for H3K27ac). Prior to library preparation, ChIP-qPCR validations were performed using primer pairs of known active and repressed genes in human LCLs (active: CEP250, GAPDH, Beta-Actin; repressed: Sox2) and a negative intergenic region as a control for the histone modifications studied. About 2–10 ng of ChIP DNA were used to prepare the sequencing libraries using the NEBNext Ultra DNA Library Prep Kit for Illumina (NEB, E7370L) as per the manufacturer's instructions. ChIP-seq libraries were size selected to remove fragments below 100 bp and amplified for 10 PCR cycles. Four to five libraries were pooled and loaded into a total of 13 lanes. About 50 bp single-end reads (50SE) were sequenced in an Illumina HiSeq2500 machine, using v4 chemistry.

**Assay of transposable chromatin**. ATAC-seq libraries for nine lymphoblast cell lines were generated following the Buenrostro et al. protocol[79]. Briefly, 50,000 cells were harvested, washed in cold PBS and resuspended in 50 mL of cold lysis buffer (10 mM Tris-HCl pH 7.4, 10 mM NaCl, 3 mM $MgCl_2$, 0.1% (v/v) Igepal CA-630). Samples were spun down for 10 min at $500 \times g$, 4 °C. Pellets were resuspended in the transposition reaction mix and incubated at 37 °C for 30 min. Samples were purified using the Qiagen MinElute PCR purification kit. Transposed DNA was eluted in 10 µL of elution buffer and subjected to PCR amplification for eight cycles using barcoded primers and NEBNext High Fidelity PCR master mix. ATAC-seq libraries were purified using 1.8X volumes of AMPure XP beads to remove fragments below 100 bp. Library quality was assessed using a Bioanalyser High Sensitivity DNA analysis kit (Agilent). All nine libraries were pooled and loaded in a total of three lanes. 50 bp paired-end reads (50PE) were sequenced on a HiSeq 2500 platform (Illumina), using v4 chemistry.

**Definition of RE**. This analysis was performed to identify putative RE in each primate species. We used ChromHMM to jointly learn chromatin states across samples and segment the genome of each sample[26]. ChromHMM implements a multivariate Hidden Markov Model aiming to summarize the combinatorial interactions between multiple chromatin datasets. Bam files from the five histone modifications profiled were binarized into 200 bp density maps. Each bin was discretized in two levels, 0 or 1, depending on their enrichment computed by comparing immunoprecipitated (IP) versus background noise (input) signal within each bin and using a Poisson distribution. Binarization was performed using the BinarizeBam function of the ChromHMM software[26]. A common model across species was learned with the LearnModel ChromHMM function for the concatenated genomes of all samples but O1 (orangutan sample 1) due to its anomalous methylation profile (Supplementary Fig. 43). Several models were trained with a number of chromatin states ranging from 8 to 20. To evaluate the different *n*-state models, for every sample, the overlap and neighborhood enrichments of each state in a series of functional annotations were explored. A 16-state model was selected for further analysis based on the resolution provided by the defined chromatin states, which capture the most significant interactions between histone marks and the state enrichments in function-annotated datasets (Supplementary Fig. 2). The genomic coordinates of RE were defined for each sample by merging all consecutive 200 bp bins excluding elongating (E1 and E2), repressed heterochromatin (E16), and low signal (E15) chromatin states. Species RE were defined as the union of sample RE. For orangutan, we did not include RE specific to O1.

**Assignment of a regulatory state to RE**. This analysis was performed to assign each RE a regulatory state based on their epigenetic signals. We did so using a two-step approach. First, we classified RE based on the chromatin states found in them. Second, we refined this classification based on their underlying histone modifications and open chromatin signals.

RE were assigned a chromatin-state based annotation. Combining the information gathered through the overlap and neighborhood enrichment analyses in functionally defined regions, we established a hierarchy to designate poised (p), strong (s), and weak (w) promoter and enhancer states. Chromatin states E8, E9, and E11 defined promoter states (P); E8 and E9 were strongly enriched at TSSs, CGI, UMR (unmethylated regions), and open chromatin regions, while E11 was mostly located downstream the TSS; the presence of E14 defined poised promoter states (pP); absence of E14 and presence of E9 or E11 defined strong promoter states (sP); remaining P were classified as weak promoter states (wP). Non-promoter RE were assigned an enhancer state (E). The presence of E14 defined poised enhancer states (pE); absence of E14 and presence of E3, E4, E5, E6, and E12 defined strong enhancer states (sE): E5 and E6 were strongly enriched LMRs (low methylated regions) whereas E3, E4, and E12 were highly abundant at introns; remaining E were classified as weak enhancer states (wE) (Supplementary Figs. 2 and 33).

One of the limitations of chromatin states is that bin assignments are based on the presence or absence of particular epigenetic marks. However, oftentimes, the lines separating different RE are blurry: e.g., the distinction between promoter and enhancer states generally resides in the H3K4me3/H3K4me1 balance. Hence, some

misclassifications are expected due to an insufficient precision of the qualitative classification. Considering the quantitative relationship between co-existing histone modifications can help to accurately annotate epigenetic states in RE. We used linear discriminant analysis (LDA)[80] to refine chromatin-state based annotations. This method is commonly applied to pattern recognition and category prediction. LDA is a technique developed to transform the features into a lower-dimensional space, which maximizes the ratio of between-class variance to the within-class variance, thereby granting maximum class separation. We performed LDA analysis using the lda function in the R package MASS (version 7.3-47)[81]. The predictor variables were the background-noise normalized IP signals from the five different histone modifications profiled and the chromatin accessibility signal at species RE. The categorical variable to be predicted based on the underlying enrichments was the chromatin-state-based annotation. The regulatory state at the species level was determined based on the regulatory state in each of the biological replicates. Thus, the regulatory state of an element with different epigenetic states in the two replicates (ambiguous) could be aP or aE, when both samples of a given species were annotated as either P or E but differ in their activity; P/E, when a RE was classified as P in one biological replicate and E in the other one; and P/Non-RE or E/Non-RE, when the RE was detected only in one replicate (Supplementary Fig. 6 and Supplementary Data 1). Larger inter-sample variability is expected between human replicates as H1 data were obtained from previously published datasets. Hence, to control for interindividual variability, only RE with the same activity in the two replicates were considered for downstream analyses.

**Analysis of evolutionary conservation at orthologous regulatory regions**. This analysis was performed to explore the conservation patterns of RE across species. We studied the evolutionary conservation of epigenetic signals at two levels. First, we investigated the conservation of RE with promoter or enhancer epigenetic states, regardless of their activity. Second, we investigated the evolutionary conservation of strong, poised, and weak promoters and enhancers.

To study the evolutionary conservation of promoters and enhancers, we focused on a set of 21,753 one-to-one orthologous regions associated with genes, in which at least one species showed a promoter or enhancer epigenetic state. To measure the evolutionary conservation at each orthologous regulatory region, we count the number of species (1, 2, 3, 4, or 5) in which the promoter or enhancer state is conserved (Supplementary Fig. 9).

To study how evolutionary changes occur, we define recently repurposed promoters as orthologous regulatory regions in which one species shows a promoter state while the others show an enhancer state or vice versa. Similarly, we refer as novel promoter or enhancer states to those orthologous regulatory regions in which a given species showed a promoter or enhancer state while the others showed no evidence of regulatory activity (classified as nonregulatory).

To study the evolutionary conservation of strong, poised, and weak promoters and enhancers, we focused on the subset of 10,641 one-to-one orthologous regions in which at least one species showed a strong, poised, or weak regulatory state (we do not include orthologous regions including elements with ambiguities, i.e., different activities between biological replicates). To measure the evolutionary conservation at each orthologous regulatory region, we count the number of species (1, 2, 3, 4, or 5) in which each regulatory state (strong, poised, and weak promoters or enhancers) is conserved (Supplementary Fig. 12). To statistically assess the different evolutionary dynamics for the different regulatory states, we first ran randomization analyses. We randomized (1000 randomizations) the regulatory states associated with each species in the orthologous regulatory regions. We calculate the P value as the number of randomizations with average conservation equal to or above the observed conservation for each regulatory state. We further explored the different patterns of conservation combining: (1) Kruskal–Wallis test (kruskal.test R function)[82] to test whether the global distributions of the number of species in which each particular state was conserved were different for the different regulatory states, (2) Dwass–Steel–Critchlow–Fligner test to assess the significance of every pairwise comparison (dscfAllPairsTest function from the R package PMCMRplus version 1.4.4)[83], and (3) Glass rank biserial correlation coefficient for Mann–Whitney U-test to compute the effect sizes associated with all statistically significant pairwise comparisons (wilcoxonRG function from the R package rcompanion version 2.3.25)[84].

To study the patterns of evolutionary conservation of the sequence underlying orthologous regulatory regions, we assigned each orthologous regulatory region a conservation score. We computed this score based on the phastCons30way sequence conservation track[29]. To control for background sequence conservation levels, we first computed the average and standard deviations phastCons30way in TADs defined in the cell line GM12878[85] (Supplementary Fig. 13). Then, we used these summary statistics to calculate the Z-score for each bp in every orthologous regulatory region, using the average and standard deviations values of the TAD in which each orthologous regulatory region was found. We averaged the Z-scores within each orthologous regulatory region in bins of 200 bp that overlap 50 bp with the next bin and assign each orthologous regulatory region the maximum Z-score values associated with its bins. We computed the Spearman rho correlation between the Z-scores and the number of species in which each orthologous regulatory region was conserved, separately for each regulatory state. To determine the statistical significance of these correlations we used randomization analysis. For

each regulatory state, we created 1000 sets randomizing the $Z$-score associated with each orthologous regulatory region and calculated the Spearman correlation in each randomization. We determined the $P$ value as the number of randomizations with a Spearman rho correlation value equal to or above the observed correlation (Supplementary Fig. 15).

**Classification of RE in different types of components of gene regulatory architectures**. This analysis was performed to classify RE based on their putative role in the regulation of gene expression.

We pre-classified each RE into a gene regulatory component based on their genomic location with respect to their corresponding species ENSEMBL release 91[86] gene annotations. RE found up to 5 Kb upstream to the nearest TSS were classified as gP. Additional RE located up to 10 Kb to the nearest TSS were classified as prE. RE that overlapped a gene were classified as gE. Other RE that could not be linked to a gene based on their genomic proximity were initially classified as distal enhancers (dE).

Then, we made use of available interaction data for the cell line GM12878 (HiC[31], HiChIP-H3K27ac[32], and ChIA-PET[33]) to map interactions between RE. Each interacting pair was mapped independently to hg38 coordinates using the liftOver tool from the UCSCTOOLS/331 suite[87], and only interactions for which both pairs could be mapped were kept. Subsequently, interactions were mapped to the nonhuman primate reference genome assemblies. For interspecies mappings, coordinates were mapped twice, going forward and backward, and only pairs that could be mapped in both directions were kept. Interacting RE were defined as those that overlapped with each pair of any given interaction. First-order interactions were annotated between promoters and enhancers, allowing the definition of PiE. Second-order interactions were annotated between enhancer components (gE, prE, or PiE), allowing the definition of EiE (Fig. 4a and Supplementary Fig. 1).

Considering both their epigenetic state and regulatory component, RE were separated into 30 ($6 \times 5$) different subcategories. We used a Chi-square test to identify the component-epigenetic state combinations enriched in orthologous regulatory regions with fully conserved and species-specific epigenetic states (Supplementary Fig. 26).

**Gene expression levels and regulatory states in gene components**. This analysis was performed to explore whether gene expression levels are associated with the activity state of RE in each type of component.

To investigate the influence of the activity state of RE in each type of component on gene expression levels, we classified one-to-one orthologous protein-coding genes, separately for each species, into six mutually excluding categories, one for each regulatory state within each type of component (component-state combinations). Whereas genes can only be associated with one genic promoter and hence, they can only be classified into one category for gP depending on the corresponding epigenetic state of the RE, genes can be associated with more than enhancer component (gE, prE, PiE, and EiE). In those cases, we classified genes into a given component-state category accordingly to the presence of at least one RE with a given epigenetic state in that component using the following state hierarchy: pE > pP > sE > sP > wE > wP (Supplementary Fig. 20). To statistically assess the influence of each state in each component, we used (1) Kruskal–Wallis test (kruskal.test function as implemented in R)[82] to test whether the distributions of the expression levels of genes associated with each component-state combination were different for the different regulatory states, (2) Dwass–Steel–Critchlow–Fligner test to assess the significance of every pairwise comparison (dscfAllPairsTest function from the R package PCMRplus version 1.4.4)[83], and (3) Glass rank biserial correlation coefficient effect size for Mann–Whitney $U$-test to compute the effect sizes associated with all statistically significant pairwise comparisons (wilcoxonRG function from the R package rcompanion version 2.3.25)[84] (Supplementary Fig. 21).

**Partial correlation analysis**. This analysis was performed to understand in greater detail how the epigenetic signals associated with RE with different epigenetic states and in different components relate to each other and with gene expression.

To disentangle the network of direct co-dependencies between the different components, regulatory states, histone marks, and gene expression, we performed a series of partial correlation analyses[34,88]. To tackle the diversity of architectures detected for the different genes, we added up the calibrated signal of all the RE with a given regulatory state (promoter or enhancer) in a given type of component for any given architecture. This decision was based on the observed relationship between the number of strong elements in a gene architecture and the expression level of its target gene. Separation of histone signals in each type of component between those contributing to a promoter or to an enhancer was intended to reflect the potential differences in their role in gene expression regulation. As a result of this design, our system has 51 variables (RNA-seq signal + 5 histone mark signals x 2 regulatory states x 5 components) and 57,370 cases (5737 genes x 5 species x 2 samples).

All partial correlation analyses were performed using an adaptation of a recently published Sparse Partial Correlation Analysis protocol[34] based on the continuous values of the accumulated ChIP-seq signals (instead of their ranks) to take advantage of their pseudo-quantitative nature. This protocol combines the recovery

of statistically significant partial correlations with a cross-validation process to filter out those relationships leading to overfitted reciprocal linear LASSO models (significant partial correlations unlikely to be biologically meaningful). In our case, in every analysis, we recovered those partial correlations recovered in at least four of the five species without leading to overfitting when determining the reciprocal explanatory power in the remaining species. This protocol is intended to detect biologically relevant co-dependences out of the set of significant partial correlations and as a result, this approach filters out many significant partial correlations with very low explanatory power. In fact, all the partial correlations recovered in any of the analyses performed showed very low $P$ values (Benjamini–Hochberg's correction[89], $P < 1.8 \times 10^{-22}$). In our case, given the relatively small amount of data, we focused on recovering those partial correlations that are likely to be relevant in any species. For these analyses, we used a modified version of the R code provided by the authors (http://spcn.molgen.mpg.de/code/sparse_pcor.R/) to perform fivefold cross-validation analyses separating by species instead of the original tenfold cross-validation protocol suitable for larger datasets. Network visualizations were performed with Cytoscape[90].

In a partial correlation model, direct co-dependencies are established between individual variables. However, we know that coordination of the different histone marks within components is important to define the global epigenetic configuration of a component (also captured in our epigenetic states), which itself could be considered the relevant variable for this analysis. To better address this situation in our analysis, we defined a consensus signal for every component following the same approach established by WGCN[35] to define eigengenes as representative variables of clusters of co-expressed genes. In brief, we defined eigencomponents as the variables summarizing the common signals of the different histone marks in a component (actually calculated as the first PCA component of these five variables). So that eigencomponents keep the meaning of the activities, they were defined as codirectional with H3K27ac signals in each component (eigenvectors negatively correlated with H3K27ac signals were multiplied by -1). We performed a Sparse Partial Correlation Analysis of these ten eigencomponents and RNA-seq that recovers very clearly the structure of direct co-dependencies between the epigenetic configuration of the different components and gene expression (Fig. 4e and Supplementary Data 5).

In addition, we defined the remaining unexplained signal of every histone mark by its eigencomponent as the residuals of a linear model of the original variables and the corresponding eigencomponent. A Sparse Partial Correlation Analysis of these residuals (Supplementary Fig. 22 and Supplementary Data 6) shows that even these residuals reflect the same inter-component structure and highlights that our eigencomponents miss some relevant information for the definition of this regulatory coordination (mainly weaker co-dependencies involving promoter states in intragenic and PiE and enhancer states in promoters).

To assess to what extent eigencomponents reflect the behavior of the whole network of co-dependencies of the histone marks or of each of the specific histone marks, we also performed SPCAs using the actual ChIP-seq enrichment signals. A global partial correlation analysis considering all 51 variables shows a very clear structure of direct co-dependencies with a strong intra-component contribution for the two states of every single component and a clear but most modest exclusive inter-component contribution (Supplementary Data 7). Analyses to determine the Sparse Partial Correlation Network of each of the histone marks and RNA-seq without considering the possible influence of the remaining histone marks (Supplementary Data 8) retrieve very similar networks pointing to the common backbone of inter-component co-dependences reflected in our SPCA of the eigencomponents.

Our dataset of regulatory components shows a quite unbalanced contribution of the components to the architectures, with gE being the most abundant type of component and promoter-interacting and EiE being the least abundant (Supplementary Fig. 18). These differences could be at least partially related to our inability to recover some of the chromatin interaction-mediated regulatory associations. More importantly, this imbalance, if not real, could affect the ability of our partial correlation networks to reflect the contribution of those components less represented in our datasets. To explore this point, we recovered the subset of genes (an average of 1068 genes per sample) with full architectures (those with at least one element in every type of component) and repeated all the Sparse Partial Correlation Analyses explained above with this dataset of genes. In all the cases, we obtained very similar results, recovering fewer relevant partial correlations due to the smaller number of genes, but with no signal of any relevant difference in the global structure of the coordinated network of components and gene expression (Supplementary Fig. 23 and Supplementary Data 5–8).

All the components of the connected network can be very influential in gene expression through their direct or indirect connection with gene expression. However, our Sparse Partial Correlation Networks point consistently to the direct co-dependency of RNA-seq with the genic promoter and gE components and the co-dependency between them. To quantify the explanatory power of these dependencies for gene expression, we performed a simple generalized linear model (glm function as implemented in R)[82] for RNA-seq using H3K27ac, H3K27me3, and H3K36me3 signals in gP and gE and the interactions between them. This model was able to explain 67% of the gene expression variance (Supplementary Data 8), a percentage 5% higher than the 62% explained by a naïve model including the signals of all histone marks in all the components but no interaction between them (Supplementary Table 8), supporting that gP and gE contained

nearly all the epigenetic information needed to define gene expression levels in our data.

**Differential gene expression analyses**. This analysis was performed to find genes with expression changes between species and investigate how these expression changes are connected to changes in gene regulatory components with different epigenetic states.

We identified genes with differential expression levels across species using the iDEGES/edgeR pipeline in the R package TCC (version 1.12.1)[91,92] at an FDR of 0.1 and testing all species pairwise comparisons. Then, we determined the patterns of differential expression, species, and direction of the gene expression change, using a two-step approach. For every gene, the $Q$-values obtained in species pairwise comparisons were ordered from smallest to largest. Different expression labels were then assigned to each species according to the ordered $Q$-values. Once all species had an assigned label, the average normalized expression values between groups were compared to determine the directionality of the change. We separate differentially expressed genes into two categories: genes with species-specific expression changes and genes with non-species-specific expression changes.

To investigate the relationship between changes in gene expression and changes in the regulatory architecture of a gene, for every type of regulatory component, we run a Wilcoxon signed-rank test evaluating whether the number of RE with a given regulatory state in that particular regulatory component was significantly associated with higher expression levels, for strong and weak activities, or lower expression levels, for poised activities. $P$ values obtained for each regulatory role were corrected for multiple testing using the Benjamini–Hochberg procedure[89].

**Over-representation analyses (ORA) of functional annotations**. This analysis was performed to explore the functional role of both conserved as well as species-specific regulatory states in certain regulatory/gene components (overrepresented combinations of regulatory states in specific components, Supplementary Fig. 26). For that, we defined sets of genes associated with fully conserved and species-specific component-epigenetic state combinations and explored their functional enrichments. To ensure the representativeness of the functional enrichments, for the gene lists associated with each type of component, we excluded genes associated with components with different epigenetic states activities (i.e., genes associated with both conserved strong and weak gE) or associated with both conserved and species-specific components levels (i.e., genes associated with both a conserved and a species-specific weak gE) and kept those gene lists with a minimum of 15 genes for enrichment analyses. Of note, orangutan-specific component-epigenetic state combinations were excluded from the analysis because they were defined using only one LCL replicate (see above) and they are likely to be enriched in inter-individual variable activities.

Over-representation of Gene Ontology (GO) terms was performed using the WebGestaltR function from the R package WebGestaltR (version 0.4.3; http://www. webgestalt.org) with minNum = 25 and remaining default options. This function controls the FDR by applying Benjamini–Hochberg procedure (default threshold FDR = 0.05)[89,93]. Previous analyses have shown that recent enhancers tend to occur in the same genes that already have highly conserved enhancers[10]. To control for the particular background of each component, we built different background gene sets, including the set of human genes associated with at least one-to-one orthologous regulatory regions of each type of component. This way, we have specific and different backgrounds for gP, gE, and PiE. To represent and compare enriched GO terms between component-state combinations, we performed a clustering analysis of all significantly enriched GO terms using REVIGO[94]. We associated each GO term with the proportion of genes from each component-state combination that overlapped that GO term. In the case of GO terms enriched in more than one gene set, we chose the highest proportion of genes. We used this list as input for REVIGO. Given that REVIGO only reports the clustering of ~350 GO terms and our input list was larger than that, we used the R package GofuncR (version 1.8.0)[95] to retrieve the parent GO terms of the remaining unassigned GO terms and add them to the corresponding group as defined by REVIGO. REVIGO group names were manually assigned, taking into account the most representative parent term (Supplementary Data 10).

**Analysis of tissue-specific gene expression**. The aim of this analysis was to inspect the expression pattern and tissue-specificity of certain RE in particular components (both conserved as well as species-specific) across different human cell types or tissues. We defined sets of human genes associated with fully conserved component-state combinations and human genes associated with human-specific gains/losses of RE. Note that these gene lists are not mutually exclusive since a gene can be associated with different types of conserved or species-specific component-state combinations (e.g., a gene with both a human-specific intragenic enhancer with weak activity and a fully conserved gE with a strong activity). We obtained expression levels (median TPM values) across a collection of different tissues from the latest GTEx release (v8)[36]. We only included tissues with at least 70 samples and grouped tissue subregions into the same tissue category, as stated in Supplementary Table 9. For each component-state combination, we followed a two-step approach to remove consistently low-expressed genes across tissues. For that, we first assigned a value of 0 to all genes with a median expression level below 0.1

TPM and then we excluded from the analyses those genes that had an accumulated expression value in all tissues below 0.1xNumber of tissues ($n = 29$ tissues). For each component-state combination, differences in median expression across tissues were assessed with the Friedman test using the friedman.test function as implemented in R[82]. We used the Wilcoxon–Nemenyi–McDonald–Thompson test implemented in the pWNMT function of the R package NSM3 (version 1.14)[96] to assess whether expression levels were significantly different for all pairwise tissue combinations. Then, we made use of the rank-biserial correlation to calculate the effect sizes for all statistically significant pairwise tests with the wilcoxonPairedRC function of the R package rcompanion (version 2.3.25)[84].

We then evaluated the tissue-specificity of the genes associated with the different component-state combinations. For this, we calculated the tissue specificity index[97] ($\tau$, tau) for each gene, which is defined as:

$$\tau = \sum_{i=1}^{N} (1 - x_i)/N - 1 \qquad (1)$$

where $N$ is the number of tissues and $x_i$ is the expression value normalized by the maximum expression value. This value ranges from 0, for housekeeping genes, to 1, for tissue-specific genes (values above 0.8 are used to identify tissue-specific genes)[98]. Tissue-specificity indices were calculated for all genes included in the latest GTEx release[36]. Gene expression levels (median TMP) of grouped tissue categories (Supplementary Table 9) were normalized within and across tissues before calculating $\tau$ as implemented in the R package tispec (version 0.99.0)[99]. The calcTau function from this package provides a tau value for each gene and also a tau expression fraction for each tissue (which also ranges from 0 to 1) that indicates the specificity of a given gene for that tissue.

After calculating $\tau$ values, we compared their distributions between gene datasets with the Kruskall–Wallis test and assessed the significance of every pairwise comparison with the Dwass–Steel–Critchlow–Fligner test (dscfAllPairsTest function from the R package PMCMRplus version 1.4.4)[83]. Glass rank biserial correlation coefficient was used to compute the effect sizes associated with all statistically significant pairwise comparisons using the wilcoxonRG function from the R package rcompanion version 2.3.25[84] ($P < 0.05$).

**Association of genes containing gE with signals of positive selection and accelerated evolution**. The main objective of this analysis was to examine whether hswEge-containing genes had been targeted by different evolutionary forces. We built a database of human genomic regions with previously detected signals of positive selection in humans[50–54] and human acceleration[55–57]. In brief, for the signals of positive selection, we retrieved Supplemental Files 1 and 2 from Peyrégne[50] et al. Table S5 from Racimo[51] et al. Table S3 (regions >1600 generations) from Zhou[52] et al. Table S19a.3 from Prüfer[53] et al. and Table S37 from Green[54] et al. For the datasets with different signals of human acceleration we took haDHSs (Supplemental Table 2) from Gittelman[55] et al. HARs (Supplementary Info) from Lindblad-Toh[56] et al., and CNSs (Table S1) from Prabhakar[57] et al. To account for the potential effect that gBGC could have in these regions, we filtered out all those regions overlapping with gBGC tracts in the human genome[100]. After this filtering, BEDtools[101] was used to assign these regions to both protein-coding and non-coding genes following similar criteria to those used for building the gene regulatory architectures (Methods' section Classification of RE in different types of components of gene regulatory architectures). We assigned these regions to a protein-coding gene if they were located within the gene or up to 5 Kb upstream of its TSS. Then, we made use of available interaction data for the cell line GM12878 (HiC[31], HiChIP-H3K27ac[32], and ChIA-PET[33]) to assign this composite of positively selected and acceleration regions to their interacting genes. We ended up with a set of 4747 genes associated with at least one positively selected region or one accelerated region in the human lineage. We computed the overlaps between this gene list and the lists of genes associated with the different component-state combinations at different degrees and backgrounds. We used one-tailed Fisher's exact test to assess the enrichment significance.

**Analyses of the density of human-fixed single nucleotide changes (hSNCs) in gE with weak enhancer states**. The purpose of this analysis was to assess the co-occurrence of human-nucleotide changes in human-specific RE. In order to study the distribution of these human-fixed changes, we first generated a dataset with human-specific changes. We used sequencing data from a diversity panel of 27 orangutans, 42 gorillas, 11 bonobos, and 61 chimpanzees[102–104], as well as 19 modern humans from the 1000 genomes project[105], all mapped to the human reference assembly hg19. We applied a basic filter for each site in each individual (sequencing coverage >3 and <100) and kept sites where at least half of the individuals in a given species had sufficient data. Furthermore, at least 90% of the kept individuals at a given site in a given species had to share the same allele; otherwise, the site was labeled as polymorphic in the population. Indels and trialleleic sites were removed, and only biallelic sites were kept. We used data from a macaque diversity panel[106], applying the same filters described above. The allele at monomorphic sites was added using bedtools getfasta[101] from the macaque reference genome rheMac8. Since this panel uses the macaque reference genome, we performed a liftover to hg19 using the R package rtracklayer[107] and merged the data with the great ape diversity panel.

Lineage-specific changes were retrieved as polymorphisms with sufficient information. Hence, human-specific changes (hSNCs) were defined as positions where each species carry only or mostly one allele within their respective population, the majority of individuals in each population have a genotype call at sufficient coverage, and the human allele differs from the allele in the other populations.

BEDtools[101] was used to annotate those hSNCs in conserved or hswEgE, and the density of changes was calculated as the number of hSNCs present in each enhancer divided by the length of the enhancer.

To determine which hswEgE were enriched in human-specific changes, we compared their density to what would be expected at random. For that, we first established the number of hSNCs that fall in human gE with weak enhancer states associated with one-to-one orthologous regulatory regions (our universe of enhancers). In each simulation, this number of mutations was randomly placed in this universe, and we computed the density for each of the hswEgE (10,000 simulations). With this approach, we corrected for the differences in the length of the enhancers. The $P$ value for each enhancer was computed as the number of simulations with a density equal to or above the observed density for that particular enhancer. All $P$ values were corrected by multiple testing using the Bonferroni method with the number of tests equal to the number of hswEgE.

We then assessed whether the number of enhancers that were statistically enriched in hSNCs (or number of hits) was greater than what would be expected at random. In order to do that, for each enhancer, we defined its mutation density critical value adjusting by multiple testing and using the simulated values. For example, in a hypothetical case of 100 enhancers and 10,000 simulations, for each enhancer, we would order its simulated density of hSNCs from smallest to largest and take the fifth value as the critical one (given that our chosen alpha equals 5%, but it has to be corrected by 100 tests; therefore it becomes 0.05%). Once we established a critical value for each human-specific intragenic weak enhancer, we determined, for each simulation, how many enhancers had a density equal to or above their corresponding critical value. Finally, we computed the $P$ value comparing the number of artificial hits in each simulation with the number of observed hits.

## Data availability

The raw fastq files from the genomic, transcriptomic, and epigenomic data generated and used for the analyses in this study were uploaded to the Sequence Read Archive (SRA) with the BioProject accession number PRJNA563344. Data from human cell line GM12878 used in the study was obtained from SRA: RNA-seq accessions SRR998197 and SRR998198, ChIP-seq for H3K4me3 accession SRR998194, H3K4me1 accession SRR998191, H3K27ac accession SRR998178, H3K36me3 accession SRR998187, and H3K27me3 accession SRR998189; GM12878 ChIP-seq input accession and ATAC-seq data accession. Interaction data available for GM12878: HiC accession GSE63525, HiChIP-H3K27ac, and ChIA-PET accession GSM1872887. Expression data is available from GTEx v8. The remaining data are available within the manuscript, Supplementary Information, Supplementary Data or from the corresponding authors upon reasonable request.

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

## Acknowledgements

R.G.-P. was supported by a fellowship from MICINN (FPU13/01823). P.E.-C. was supported by a Formació de Personal Investigador fellowship from Generalitat de Catalunya (FI_B00122). M.K. was supported by a Deutsche Forschungsgemeinschaft (DFG) fellowship (KU 3467/1-1) and the Postdoctoral Junior Leader Fellowship Program from "la Caixa" Banking Foundation (LCF/BQ/PR19/11700002). D.J. was supported by a Juan de la Cierva fellowship (FJCI2016-29558) from MICINN. T.M-B. is supported by funding from the European Research Council (ERC) under the European Union's Horizon 2020 research and innovation program (grant agreement EC-H2020-ERC-CoG-ApeGenomeDiversity-864203), BFU2017-86471-P (AEI/FEDER, UE), "Unidad de Excelencia María de Maeztu", funded by the AEI (CEX2018-000792-M), Howard Hughes International Early Career, NIH 1R01HG010898-01A1, Obra Social "La Caixa" and Secretaria d'Universitats i Recerca and CERCA Program del Departament d'Economia i Coneixement de la Generalitat de Catalunya (GRC 2017 SGR 880). G.M., V.D.C., and L.D.C. were supported by grants from the Spanish of Economy, Industry, and Competitiveness (MEIC) (BFU2016-75008-P) and G.M. was also supported by the "Convocatoria de Ayudas Fundación BBVA a Investigadores, Innovadores y Creadores Culturales". J.L.G.-S. was supported by the Spanish government (grants BFU2016-74961-P), an institutional grant Unidad de Excelencia María de Maeztu (MDM-2016-0687) and the European Research Council (ERC) under the European Union's Horizon 2020 research and innovation programme (grant agreement No 740041). A.N. was supported by Fondo Europeo de Desarrollo Regional (FEDER) with project grants BFU2016-77961-P and PGC2018-101927-B-I00 and by the Spanish National Institute of Bioinformatics (PT17/0009/0020). Figures 1a, 2a, 4a, 6b, and Supplementary Fig. 1 were fully or partially created using BioRender.com. We dedicate this manuscript to the memory of Jose Luis Gomez Skarmeta: a leader, an inspiration, and a friend.

## Author contributions

T.M.-B. and J.L.G.-S. conceived the study; D.J. designed and supervised the analyses; L.D.C. supervised the work of G.M. and V.D.C.; A.B. procured nonhuman great ape cell lines; A.N. provided helpful insights; R.G.-P., G.M., and V.D.C. performed the experimental work; R.G.-P., P.E.-C., D.J., I.L., M.R., and M.K analyzed the data; D.J., R.G.-P., and P.E.-C. wrote the manuscript with input from all co-authors.

## Competing interests

The authors declare no competing interests.
