## [Peer Review File · Nature Communications]

REVIEWER COMMENTS

Reviewer #1 (Remarks to the Author):

Authors study the evolution of epigenetic marks and sequence associated with gene regulatory activities in lymphoblastoid cell lines. They use ChIP-seq for five histone marks, ATAC-seq and RNA-seq, whole genome sequencing and whole genome bisulfite sequencing in two biological independent cell lines in five close related primate species. The data allows for a comparison of the types of regulatory regions (promoters and enhancers), their location respect to the regulated gene, their state strong, poised or weak and the sequence conservation related to this cellular model system. This is interesting and complete work that reveals that comparative genomics and epigenomics are very powerful approaches to the study of the evolution of gene regulation. There are several novel results: authors reveal the importance of strong promoters and intragenic enhancers for the expression of genes in that cell type and the fact that the changes in regulation occur for weak regulatory elements that regulate tissue specific genes that evolve fast/under positive selection. I have a few comments/suggestions for the authors.

Main comments:

1. I think authors need to spell out LCLs to have a clearer title: “Epigenomic profiling of primate lymphoblastoid cell lines reveals the evolutionary patterns of epigenetic activities in gene regulatory architectures”
2. Authors say that “ Despite the artificial nature of our cellular model^{46–48}, previous studies have shown the value of LCLs as an experimentally convenient model of somatic cells that accurately resembles the phenotype of its cell type of origin⁴⁹ and which can be robustly used for comparative studies in humans and primates^{12,50–52}.” I agree that it can be a good system to understand the evolution of gene regulation in that cell type. Authors, however, are able to make a lot more general claims: “The consistent association of epigenetic and sequence conservation also suggests that the epigenetic conservation observed in LCLs is a good proxy for the conservation of the regulatory activity of these elements in our primate species.” I wonder why that is. Are the main patterns between cell types the same? Are the approaches leading to a comparison of a lot of housekeeping genes and a big fraction of other genes (tissue specific genes that evolve fast/under positive selection) have a similar behavior between most other cell types? Does the author’s approach miss some aspects of gene regulation by focusing on genes that have one to one orthologs between species?
3. If I understand correctly, ChromHMM is a software that allows learning about marks that characterize chromatin states. Have the authors learned anything new about regulatory region states and marks from their LCLs analyses? Have they confirmed any previous observation from their LCLs analyses?
4. Do authors can make predictions of how the epigenetic marks will change between cell types for tissue-specific genes? What marks are found in LCLs-specific genes?

Reviewer #2 (Remarks to the Author):

Garcia-Perez et al generated LCL lines from multiple primate species, in order to generate genomic and epigenetic data to investigate the evolution of gene expression and regulation in primates. They present interesting and potentially compelling findings supporting that the evolution of gene regulation is deeply influenced by the coordination of epigenetic activities in gene regulatory architectures. Further, they show that human-specific weak intragenic enhancers tend to be associated with brain-related functions. The paper adds important data to the state-of-the-art of the field. On the other hand, there are some issues which would need to be addressed before acceptance.

1) The manuscript is long and long-winded and very hard to read. A total of 65 supplementary figures, and an almost equal number of supplementary tables, are way too many for a paper of this size, and no reader will ever check all of them. Some of the methods are also not explained very clearly (too me at least), or not explained at all. I have read the methods multiple times, and still am not sure on how the evolutionary conservation of the enhancers and promoters was assessed. In general the method are very hard to read for someone whose background is not heavily quantitative. Given that the journal has a broad readership, this should be addressed.

2) The authors did not cite all the relevant literature. Most of the papers published in the last 5-10 years on the evolution of gene regulation/expression in primates were neither cited nor discussed by the authors. Some examples (but there are plenty more that are missing): PMID: 23675311; PMID: 28855262; PMID: 30322406. The lack of comparison with updated and relevant literature also brings up some concerns on the novelty introduced by this study. However, as a reviewer, is not my job to judge the novelty of the paper.

3) In Fig. 2B, the difference in the number of recovered human vs chimp promoters seems too high compared to recent literature. Could the author explain such difference?

4) I do not fully understand the usage of H3K36me3 for this study, and specifically to detected cis-regulatory elements. As the authors may know, H3K36me3 is co-transcriptionally deposited at gene bodies while POL2 transcribes mRNA. Notably, the promoter regions are completely devoid of H3K36me3, and so are the intergenic enhancers. This histone mark is simply a mark for elongating RNA-Pol2, but does not overlap regulatory elements, with the exception of intronic enhancers. The authors should explain the rational behind choosing this specific histone modification.

5) There is nothing in the method, not even a specific section, on how the LCL panels where generated. Were they generated in batches including all species? If not, how did they account for batch effect? Similarly, how did they organize the RNA-seq ATAC-seq and CHIP-seq batches? Did they control for species-specific/batch-specific effects? There is nothing at all in the methods on how the data were generated.

6) The approach used to define weak/strong/poised CREs should be made clear in the main text and

more clearly explained in the figures. They should explain clearly and more in detail how a weak enhancer is different from a poised one, and so on.

Reviewer #3 (Remarks to the Author):

García-Pérez et al. have considered genome-wide data sets for the LCLs of 5 primate species (human, chimpanzee, gorilla, orangutan, and macaque), namely 5 histone marks (H3K4me1/3, H3K36me3, H3K27ac/me3), ATAC-seq, WGS, WGBS and RNA-seq. These data sets are extensive, and will be of value to others. The diverse analyses appear to have been performed to a high level. The study's results are consistent with what exists in the literature for more divergent species sets (e.g. repurposing of regulatory elements ref 18; Enhancer evolution across mammals ref 7; gene expression evolution ref 9). The two aspects that most caught-my-eye were: (a) that the 15 variable GLM explained two-thirds of gene expression variation, and (b) the discussion about positive selection of DNA variants in regulatory elements.

Major comments.

(1) The analysis of positive selection is the weakest aspect of this study. It appears that the gene that are being considered as being under “positive selection” are from refs 27-29. These publications discuss “accelerated evolution” (rather than positive selection) whose rationale could be GC-biased gene conversion (a phenomenon associated with recombination) and other mutational biases. Indeed, are the three human-specific intragenic enhancers that accumulate more hSNCs than expected altered by GC-biased gene conversion? If so, then the substituting base would be biased towards G or C (Glemin et al. *Genome Res* 25(8): 1215). Furthermore, rapid evolution can be caused by relaxed purifying selection and this needs to be acknowledged where relevant, perhaps including where “the hSNCs density is higher in human-specific intragenic enhancers”. Rather than using these gene lists which are susceptible to alternative conclusions (e.g. FOXP2, Atkinson et al. *Cell* 174, 1424), the authors should use a more up-to-date data set, for example the genome-wide S-score generated by Green et al. (*Science* (2010) 328, 710) that made use of the Neandertal genome sequence.

(2) The authors should take extra care not to over-interpret their epigenetic data. For example, they argue that “These results confirm that the epigenetic activities of genic promoters, intragenic enhancers, and their interactions are likely the most direct determinants of gene expression regulation in our regulatory architectures.” Nevertheless, epigenetic marks are known, in some contexts, to be downstream consequences (and not upstream causes) of genic or enhancer activities so this statement needs toning down. They should be more explicit that the elements that are annotated are only putative regulatory elements, because some will not contribute to cellular or organismal phenotypes. Similarly, they should not imply causation when only correlation is known (e.g. “The number of active enhancers located near a gene – its regulatory complexity – has also been reported to influence the conservation of gene expression in mammals”). Similarly, the partial correlation analysis (e.g. Figure 4e) is not necessarily reflective of causation and this needs to be stated explicitly.

(3) I assume that the authors will make the data sets readily available. I could not find a statement in the submission to this effect. Are all annotations being made available, e.g. PiEs and EiEs?

Minor comments.

(4) Figure 2. Define aP and aE (“ambiguous”?) in the main text /legend.

(5) “61% are 1-to-1 orthologous protein-coding genes in all primate species (Fig. 2c)” – explain that the 61% refers to REs and not genes.

(6) Typos: Suppl Figure 2 “Probabilites”; Suppl Figure 8 “Number of regulatory elements with same the”. Is it simply by chance that 4 of 5 of the Wilcoxon p-values are identical? Suppl Figures 20, 21 and 22 – should the figure legends be identical? Suppl Figure 25 “phatsCons30 scores associated with TADs defined in cell the line”. Suppl Figure 30 “sates”. Explain in their legends how Suppl Figures 38 and 39 differ. Suppl Figure 55 Wilcoxon signed rank test. Suppl Figure 62, confirm that X-axis is hSNC density and Y-axis is number. Do not show smoothed data. Figure 4d: is the X-axis a fraction, and not a percentage? Line 337 “gene set shows”

(7) The authors should discuss whether there are any limitations in their approach of counting species versus using a phylogenetic approach.

(8) Noncoding gene models are likely to be most complete for human. Explain how this may change your study’s results and alter its conclusions.

(9) Line 200-202 Remove: “Note also that conserved poised promoters are associated with very high conservation z-scores, which probably did not reach significance due to their low number (n = 9 pP)” because this is speculation.

(10) Figure 5. The authors need to better explain why, e.g., these findings in LCLs (an immune cell model) are related and biologically relevant to “developmental and proliferative functions” and “neuronal functions”. The overarching conclusion (“These unexpected associations are likely to reflect the importance of particular epigenetic states in certain regulatory components to regulate specific processes”) was not particularly informative. I was not particularly convinced by this section.

Chris Ponting

REVIEWER COMMENTS

We would like to start by thanking all the reviewers for their time and positive feedback, which we found to be very helpful. We have now revised the paper and made the required changes according to the reviewer's comments, as we detail below.

Reviewer #1 (Remarks to the Author):

Authors study the evolution of epigenetic marks and sequence associated with gene regulatory activities in lymphoblastoid cell lines. They use ChIP-seq for five histone marks, ATAC-seq and RNA-seq, whole genome sequencing and whole genome bisulfite sequencing in two biological independent cell lines in five close related primate species. The data allows for a comparison of the types of regulatory regions (promoters and enhancers), their location respect to the regulated gene, their state strong, poised or weak and the sequence conservation related to this cellular model system. This is interesting and complete work that reveals that comparative genomics and epigenomics are very powerful approaches to the study of the evolution of gene regulation. There are several novel results: authors reveal the importance of strong promoters and intragenic enhancers for the expression of genes in that cell type and the fact that the changes in regulation occur for weak regulatory elements that regulate tissue specific genes that evolve fast/under positive selection. I have a few comments/suggestions for the authors.

We truly appreciate this positive comment.

Main comments:

1. I think authors need to spell out LCLs to have a clearer title: "Epigenomic profiling of primate lymphoblastoid cell lines reveals the evolutionary patterns of epigenetic activities in gene regulatory architectures".

We thank the reviewer for their suggestion. We have updated the title of the manuscript that now reads: *Epigenomic profiling of primate lymphoblastoid cell lines reveals the evolutionary patterns of epigenetic activities in gene regulatory architectures.*

2. Authors say that “Despite the artificial nature of our cellular model^{46–48}, previous studies have shown the value of LCLs as an experimentally convenient model of somatic cells that accurately resembles the phenotype of its cell type of origin⁴⁹ and which can be robustly used for comparative studies in humans and primates^{12,50–52}.” I agree that it can be a good system to understand the evolution of gene regulation in that cell type. Authors, however, are able to make a lot more general claims: “The consistent association of epigenetic and sequence conservation also suggests that the epigenetic conservation observed in LCLs is a good proxy for the conservation of the regulatory activity of these elements in our primate species.” I wonder why that is. Are the main patterns between cell types the same? Are the approaches leading to a comparison of a lot of housekeeping genes and a big fraction of other genes (tissue specific genes that evolve fast/under positive selection) have a similar behavior between most other cell types? Does the author’s approach miss some aspects of gene regulation by focusing on genes that have one to one orthologs between species?

The reviewer raised a very interesting question. As the reviewer correctly points out, our focus on one-to-one orthologies leads to an underrepresentation of tissue-specific regions as many of the regions with recent complex duplication/deletion histories belong to this class. In this sense, our analyses do not reflect the effect of evolutionary innovations implying major recent genomic changes, but deal with more subtle effects of regulatory evolution in evolutionarily comparable regions between species. However, as we have shown in our tissue-specificity analyses in humans (see Results section *Different regulatory components with poised and weak enhancer states appear in LCL-unrelated genes and functions* and Methods section *Analysis of tissue-specific gene expression*), we still include an important number of tissue-specific regulatory regions and genes.

Our focus on LCLs only allows us to see regulatory regions detectable (not only active) in these cells. However, these regions are a combination of widely active and cell type-specific elements. In fact, the definition of weak and poised activities enriches our representation of tissue-specific regions containing even regions only strongly active in other cell types. Precisely, this is what we observe in genes that harbour human-specific weak intragenic enhancers and also genes with conserved poised enhancers in gene promoters, both showing brain-specific expression patterns. In consequence, our ability to observe the interrelation between sequence and epigenetic conservation is a natural consequence of the good representation of both conserved (enriched in strong active regions and including housekeeping) and recent (tissue-specific) regulatory regions in the LCLs regulatory landscape. We have clarified this point at the end of Results section *Different regulatory components with poised and weak enhancer states appear in LCL-unrelated genes and functions* (lines 367-373):

“Our results show that conserved strong epigenetic states are involved in the regulation of important genes highly expressed in LCLs and other tissues. However, poised and weak enhancer states, either conserved or human-specific, are involved in the regulation of genes marginally expressed in LCLs, but

with particular functional roles and tissue-specific expression patterns. These unexpected associations, along with the evolutionary conservation patterns of poised and weak enhancer states, suggest that these epigenetic states might be indicative of putative regulatory elements in other cell types different from LCLs.”

3. If I understand correctly, ChromHMM is a software that allows learning about marks that characterize chromatin states. Have the authors learned anything new about regulatory region states and marks from their LCLs analyses? Have they confirmed any previous observation from their LCLs analyses?

We appreciate the reviewer's question and interest in better understanding the rationale we applied and our results.

Individual histone modifications can be studied independently, using genomic regions with significant enrichments (peaks). However, when several histone modifications are profiled, like what we do in this study, it is advised to consider the additional information gained with the combination of different epigenetic signals in their spatial context, this is, the co-occurrence of histone modifications at particular genomic regions. As an example, H3K4me3 is considered the hallmark of promoters¹ and profiling its genome-wide distribution in any given model system would determine putative transcriptional start sites (TSS). The additional profiling of H3K27ac, a hallmark of active regulatory elements², would allow the identification of active promoters. Similarly, the additional profiling of H3K27me3, hallmark of repressed/poised regulatory elements³, would allow the identification of inactive promoters. The term *chromatin state* was coined to refer to such combinations of chromatin modifications patterns⁴. Briefly, ChromHMM implements a multivariate Hidden Markov Model to summarize the combinatorial interactions between multiple chromatin datasets and generates a 200-bps density map in which each bin is discretized in two levels, 0 or 1, depending on the enrichment of each histone modification.

Here, we first used ChromHMM to identify the chromatin states that could be observed across cell lines using the five histone modifications we profiled: H3K4me3, H3K4me1, H3K27ac, H3K36me3 and H3K27me3. We chose a model with 16 states (Supplementary Fig. 2), which captures the most significant and previously described interactions between chromatin marks. Note here, that we did not intend to identify new or different associations from the ones already described in previous works, but rather to learn where in the genome of the different primate species these chromatin states appear. From this analysis, we confirmed that previously established associations between histone modifications replicate across primate species. For example, chromatin state E6 is defined by the presence of H3K4me3, H3K4me1 and H3K27ac and it is strongly enriched around annotated TSS across species (Supplementary Fig. 2). These results fall in line with previously published observations⁵ and indicate that the histone modifications regulatory patterns are conserved across mammals.

One of the limitations of chromatin states is that, as mentioned above, the bin assignments are binary, as these are based on the presence or absence of particular epigenetic signals. Yet, ChIP-seq experiments are quantitative and oftentimes the lines separating different regulatory elements are blurry: e.g., promoters and enhancers are usually marked by both H3K4me3 and H3K4me1, and the distinction between one and the other resides in the H3K4me3/H3K4me1 balance^{6,7}. Because of this, and to refine the chromatin-state based annotation of regulatory elements (Methods section *Assignment of a regulatory state to regulatory elements*) we applied a linear discriminant analysis (LDA) using the five histone modifications and the chromatin accessibility quantitative signal underlying each regulatory element and then assigned each regulatory element one out of six possible epigenetic states: active promoter, poised promoter, weak promoter, active enhancer, poised enhancer, weak enhancer (Supplementary Fig. 3). This additional step we implemented explicitly identifies, for the first time, regulatory elements with weak activities. Notably, the analyses presented in the last two sections of the manuscript highlight the functional relevance of these previously neglected classes of regulatory elements. Of note, these regulatory elements identified as weak promoters or enhancers in LCLs may probably show strong activities in different cell types.

We have now included a more detailed description of the epigenetic signals associated with each type of regulatory element in the Results section *Annotation of regulatory elements* (lines 107-113, see below) and in the caption of Fig. 2a:

Lines 107-113: “*At large, promoters have a positive H3K4me3/H3K4me1 ratio, whereas H3K4me1 is more abundant in enhancers than promoters. Regulatory elements with strong activities have an H3K27ac enrichment level similar to that of H3K4me3 in promoters and H3K4me1 in enhancers. Poised regulatory elements have a characteristic enrichment of H3K27me3. Finally, weak promoters and enhancers are associated with lower intensity enrichments of all epigenetic signals and generally lack H3K27me3 and H3K36me3 (Supplementary Fig. 3).*”

Lines 969-978 (Fig. 2a): “*Approach followed to annotate and classify regulatory elements (RE). In short, we classify RE in promoter and enhancer states with three activity levels (strong, poised or weak) based on a combination of chromatin marks and ATAC-seq signals. Bars represent relative enrichment of epigenetic signals in RE, from left to right: H3K4me3 (dark blue), H3K4me1 (light blue), H3K27ac (green), H3K36me3 (yellow), H3K27me3 (red) and open chromatin (grey). Promoters have a positive H3K4me3/H3K4me1 ratio, whereas H3K4me1 is more abundant in enhancers. Regulatory elements with strong activities are associated with high H3K27ac levels and poised regulatory elements have a robust enrichment in H3K27me3. RE are then linked to genes based on 1D gene proximity and 3D published chromatin maps for LCLs. RE not associated with any gene are referred to as orphan RE. See Methods and extended representation in Supplementary Fig. 1.*”

4. Do authors can make predictions of how the epigenetic marks will change between cell types for tissue-specific genes? What marks are found in LCLs-specific genes?

On the one hand, we expect the regulatory elements we identified in our model system, LCLs, to follow the general rules of gene regulation. Indeed, we show that regulatory elements with strong activities are associated with genes highly expressed in LCLs, whereas genes associated with regulatory elements with poised activities show low expression levels (Supplementary Figs. 20 and 21). Our functional enrichment analyses also reveal that genes with intragenic enhancers with conserved strong enhancer states are enriched in functions specific to LCLs like those involving viral processes (Supplementary Fig. 27) and show their largest expression in LCLs compared with other tissues (Supplementary Fig. 28). These results reinforce the notion that strong enhancers regulate tissue-specific expression patterns⁸, particularly intragenic strong enhancers⁹.

On the other hand, we expect the epigenetic state of different regulatory elements to change between different cell types. For example, we show that intragenic enhancers with conserved weak enhancers states in LCLs are enriched in brain-related functional annotations such as cell projection and synapse (Supplementary Fig. 27) and that the associated genes are lowly expressed in LCLs but highly expressed in brain (Supplementary Fig. 28). Hence, we hypothesize that these regulatory elements with weak activities in LCLs, if profiled in brain tissues, might be associated with strong epigenetic activities. It has also been reported that some regulatory elements may function as both promoters and enhancers¹⁰. Therefore, it could also happen that some of the promoters identified in LCLs have an enhancer-like epigenetic profile in other tissues and vice versa.

Reviewer #2 (Remarks to the Author):

Garcia-Perez et al generated LCL lines from multiple primate species, in order to generate genomic and epigenetic data to investigate the evolution of gene expression and regulation in primates. They present interesting and potentially compelling findings supporting that the evolution of gene regulation is deeply influenced by the coordination of epigenetic activities in gene regulatory architectures. Further, they show that human-specific weak intragenic enhancers tend to be associated with brain-related functions.

The paper adds important data to the state-of-the-art of the field. On the other hand, there are some issues which would need to be addressed before acceptance.

We thank the reviewer for their considerations about this manuscript.

1) The manuscript is long and long-winded and very hard to read. A total of 65 supplementary figures, and an almost equal number of supplementary tables, are way too many for a paper of this size, and no reader will ever check all of them. Some of the methods are also not explained very clearly (too me at least), or not explained at all. I have read the methods multiple times, and still am not sure on how the evolutionary conservation of the enhancers and promoters was assessed. In general the method are very hard to read for someone whose background is not heavily quantitative. Given that the journal has a broad readership, this should be addressed.

We thank the reviewer for their comment.

On the one hand, we have now revisited our Methods section and simplified the descriptions when possible. Additionally, for each section, we have incorporated an introductory sentence to state the purpose of the analysis. We hope this will ease the reader into our rationale. As an example, the Methods section *Analysis of evolutionary conservation at orthologous regulatory regions*, where we describe how we measured the evolutionary conservation of promoters and enhancers now reads (lines 582-604):

"This analysis was performed to explore the conservation patterns of regulatory elements across species. We studied the evolutionary conservation of epigenetic signals at two levels. First, we investigated the conservation of regulatory elements with promoter or enhancer epigenetic states, regardless of their activity. Second, we investigated the evolutionary conservation of strong, poised and weak promoters and enhancers.

To study the evolutionary conservation of promoters and enhancers, we focused on a set of 21,753 one-to-one orthologous regions associated with genes, in which at least one species showed a promoter or

enhancer epigenetic state. To measure the evolutionary conservation at each orthologous regulatory region, we count the number of species (1, 2, 3, 4 or 5) in which the promoter or enhancer state is conserved (Supplementary Fig. 9).

To study how evolutionary changes occur, we define recently repurposed promoters as orthologous regulatory regions in which one species shows a promoter state while the others show an enhancer state or vice versa. Similarly, we refer as novel promoter or enhancer states to those orthologous regulatory regions in which a given species showed a promoter or enhancer state while the others showed no evidence of regulatory activity (classified as non-regulatory).

To study the evolutionary conservation of strong, poised and weak promoters and enhancers, we focused on the subset of 10,641 one-to-one orthologous regions in which at least one species showed a strong, poised or weak regulatory state (we do not include orthologous regions including elements with ambiguities, i.e., different activities between biological replicates). To measure the evolutionary conservation at each orthologous regulatory region, we count the number of species (1, 2, 3, 4 or 5) in which each regulatory state (strong, poised and weak promoters or enhancers) is conserved (Supplementary Fig. 12). To statistically assess ...”

Nonetheless, we acknowledge that for a full and transparent understanding of some of the analysis we performed, a minimum background might be required. While we aim to reach the broadest audience possible, we also feel that explaining in further detail some of the approaches here implemented would unnecessarily complicate the reading of this paper. Alternatively, in these cases, we provide conceptual descriptions to help the reader understand the log of the analyses and interpret the results. Because of this, should the reader be interested in further understanding some of the methods we use, we do reference all the methods and packages used. For example, we would expect not every reader to be familiar with the partial correlation analysis we implemented here, but we do include references to previous publications that explain in further detail the approach as well as we reference the software we used.

On the other hand, the reviewer is right in pointing to our extensive Supplementary Material section. Our intention is not for the reader to go through all Supplementary Figures and Tables. Rather, we provide them for those readers interested in exploring in greater detail particular analyses or use this dataset in future studies. Although we believe that sharing figures and tables is the most direct way for readers to understand the analyses supporting our findings, we do acknowledge the Supplementary Material we provide is lengthy. Therefore, following the reviewer's advice, we have shortened our Supplementary Figures by removing non-essential figures and collapsing related ones. We now have a total of 43 Supplementary Figures, which represents a 42% reduction of the previous version of the manuscript.

2) The authors did not cite all the relevant literature. Most of the papers published in the last 5-10 years on the evolution of gene regulation/expression in primates were neither cited nor discussed by the authors. Some examples (but there are plenty more that are missing): PMID: 23675311; PMID: 28855262; PMID: 30322406. The lack of comparison with updated and relevant literature also brings up some concerns on the novelty introduced by this study. However, as a reviewer, is not my job to judge the novelty of the paper.

We thank the reviewer for their comment and agree with their observation. We have now included citations to the papers mentioned by the reviewer as well as other relevant literature. We have divided the new references into four categories;

- Comparative studies on primate/mammalian gene expression: PMID: 23258891; PMID: 24136357 and PMID: 30322406.
- Comparative studies on primate methylation: PMID: 32132541 and PMID: 31953346 (combined with gene expression).
- Studies using histone modifications in primates (combined with gene expression): PMID: 28855262; PMID: 29379187 and Castelijns et al. 2020.
- Studies on gene regulation/gene regulatory evolution in primates: PMID: 23675311 and PMID: 31323043.

We do compare our findings with previously reported results and highlight the novelty specifically contributed by our analyses. For example:

Lines 151-153: “...These results for protein-coding genes fall in line with the higher conservation of promoters previously observed in mammals¹³. In contrast, for non-coding genes, promoter states are less conserved than enhancer states...”

We state that the conservation patterns observed for promoters and enhancers associated with protein-coding genes are in concordance with previous observations. However, as a novelty, we explicitly tested separately the conservation of regulatory elements associated with protein-coding genes and non-coding genes and report that, contrary to what we and others have observed in protein-coding genes, regulatory elements with enhancer states associated with non-coding genes are more conserved than regulatory elements with promoters states.

Lines 168-173: “Our results confirm those found in more distant species^{13,27} and reinforce the generality of these evolutionary dynamics in protein-coding genes. The acquisition of regulatory states in the

regulatory architectures of non-coding genes resembles that of protein-coding genes. However, the lower conservation of promoter states associated with non-coding genes suggests that their overall higher conservation is not an intrinsic characteristic of promoter states and that it depends on their specific regulatory relevance in different genes.”

Here again, we state the similarities and differences noted between protein-coding and non-coding genes.

One of the major novelties we introduce is the classification and study of promoters and enhancers separately depending on their activity state (strong, poised or weak), explained throughout the section; “*The activity of promoter and enhancers influences their epigenetic and sequence conservation*”. This adds on previous studies which explored the conservation patterns of only active regulatory elements.

Following some examples of the novelties introduced in this paper:

Lines 189-192: “*However, strong activities in promoter states are less common for non-coding than for protein-coding genes, leading to lower conservation of promoter compared to enhancer states. This shows that differences in activity composition can lead to differences in the conservation of the regulatory architectures.*”

Lines 198-202: “*...we observed that epigenetic conservation significantly correlates with the conservation of the underlying sequence –quantified as Z-scores of background normalized PhastCons values²⁹– in all epigenetic states but weak promoter states (Fig. 3b, Methods and Supplementary Figs. 13 and 14). These correlations are seen in the architectures of protein-coding but not in non-coding genes...*”

Lines 207-208: “*The sequence conservation scores associated with strong and poised enhancers are not significantly different.*”

Lines 212-215: “*These results demonstrate that a detailed classification of promoters and enhancers with different activities into regulatory architectures provides a deeper understanding of their evolutionary constraints and dynamics, expanding previous observations in mammals¹³ that could mostly be made for active regulatory activities.*”

Finally, we also report an association of weak enhancer states to functions unrelated to LCLs (Results section “*Different regulatory components with poised and weak enhancer states appear in LCL-unrelated genes and functions*”). Our observation is based on functional enrichments but also supported by independent analyses on gene expression patterns using human gene expression data available for multiple tissues and cell types (GTEx v8). To our knowledge, this unexpected association has not been reported before, and it shows the potential that often understudied regulatory elements such as weak

and poised states might have in better understanding the complexity of gene regulation in different cell types.

3) In Fig. 2B, the difference in the number of recovered human vs chimp promoters seems too high compared to recent literature. Could the author explain such difference?

Fig. 2b shows the number of regulatory elements with the same epigenetic state (promoter or enhancer) that either have the same activity (strong, poised or weak) or different activities (aP or aE, for ambiguous promoter or enhancer) between biological replicates within each species. The difference noted by the reviewer is due to the identification of fewer weak promoters in human samples, particularly in the human biological replicate one (H1) (Author response figure 1, below). Weak elements are more likely to differ between biological replicates as they are more difficult to detect due to their subtler signals. Our quantitative approach allows us to work with them in a consistent way, but more variability is still expected for them. This variability was expected since H1 data was obtained from previously published datasets (see Supplementary Materials) which contributes to higher inter-sample differences between human biological replicates.

Author response figure 1. Number of strong, poised and weak promoters identified per biological replicate in human and chimpanzee LCLs.

We have clarified the origin of this difference in the Methods section *Assignment of a regulatory state to regulatory elements* (lines 578-581):

“Larger inter-sample variability is expected between human replicates as H1 data was obtained from previously published datasets (Supplementary Methods). Hence, to control for interindividual variability, only regulatory elements with the same activity in the two replicates were considered for downstream analyses.”

4) I do not fully understand the usage of H3K36me3 for this study, and specifically to detected cis-regulatory elements. As the authors may know, H3K36me3 is co-transcriptionally deposited at gene bodies while POL2 transcribes mRNA. Notably, the promoter regions are completely devoid of H3K36me3, and so are the intergenic enhancers. This histone mark is simply a mark for elongating RNA-Pol2, but does not overlap regulatory elements, with the exception of intronic enhancers. The authors should explain the rationale behind choosing this specific histone modification.

The reviewer is right with this point. However, we would like to clarify that we do not use H3K36me3 to detect regulatory elements. As described in the Methods section *Definition of regulatory elements*:

Lines 536-540: *“...The genomic coordinates of regulatory elements (RE) were defined for each sample by merging all consecutive 200 bp bins excluding elongating (E1 and E2), repressed heterochromatin (E16) and low signal (E15) chromatin states. Species regulatory elements were defined as the union of sample regulatory elements. For orangutan we did not include regulatory elements specific to O1”.*

So by definition, there is no regulatory element which is solely determined by the presence of H3K36me3 (chromatin states E1 and E2, Supplementary Fig. 2).

Then, as described in the Methods section *Assignment of a regulatory state to regulatory elements*, we first assign a ‘preliminary’ epigenetic state to regulatory elements (Supplementary Fig. 1) based on the presence of chromatin states E8, E9, E11 and E14 (Supplementary Fig. 2).

Lines 546-557: *“Regulatory elements were assigned a chromatin-state based annotation. Combining the information gathered through the overlap and neighborhood enrichment analyses in functionally defined regions, we established a hierarchy to designate poised (p), strong (s) and weak (w) promoter and enhancer states. Chromatin states E8, E9 and E11 defined promoter states (P); E8 and E9 were strongly enriched at TSSs, CGI, UMR (unmethylated regions) and open chromatin regions, while E11 was mostly located downstream the TSS; the presence of E14 defined poised promoter states (pP); absence of E14 and presence of E9 or E11 defined strong promoter states (sP); remaining P were classified as weak promoter states (wP). Non-promoter regulatory elements were assigned an enhancer state (E). The presence of E14 defined poised enhancer states (pE); absence of E14 and presence of E3, E4, E5, E6 and E12 defined strong enhancer states (sE): E5 and E6 were strongly enriched LMRs (low*

methylated regions) whereas E3, E4 and E12 were highly abundant at introns; remaining E were classified as weak enhancer states (wE)...”

These states are defined by the presence of H3K4me3 (E8, E9, E11 and E14), H3K27ac (E9 and E11) and H3K4me1 and H3K27me3 (E14). H3K36me3 only appears in chromatin state E11, which is mostly enriched downstream from the TSS and TES (Supplementary Fig. 2), consistent with H3K36me3 being an elongation mark.

Then, we refine the classification of regulatory elements with a linear discriminant analysis using the quantitative signal associated with the five histone modifications and accessible chromatin underlying each regulatory element (Supplementary Fig. 1). In this case, H3K36me3 signal is used in combination with the signals associated with the other histone modifications and the chromatin accessibility to quantitatively differentiate between strong, poised and weak promoters and enhancers. As expected, H3K36me3 signal is generally depleted in poised and weak promoters and enhancers (Supplementary Fig. 3), but present in strong promoters and enhancers, whose flanking regions can overlap actively transcribed regions.

5) There is nothing in the method, not even a specific section, on how the LCL panels were generated. Were they generated in batches including all species? If not, how did they account for batch effect? Similarly, how did they organize the RNA-seq ATAC-seq and CHIP-seq batches? Did they control for species-specific/batch-specific effects? There is nothing at all in the methods on how the data were generated.

We thank the reviewer for their comment and pointing out the incompleteness in these sections.

In the Supplementary Methods section *Cell line acquisition and cell growth*, we state that we did not generate the LCLs lines used in this paper. Whereas we purchased the human cell line GM19150 (H2) from the Coriell Institute, non human-great ape and macaque cell lines were kindly provided by Dr. Antoine Blancher and Dr. Gaby Dioxiadis, respectively. We have now included references to previously published papers that include in their methods information regarding how LCLs were generated. For the human cell line GM1878 (H1), we used previously published datasets, all of which are referenced in the Supplementary Methods section *Data sources for GM12878 (H1)*.

Organization of sequencing batches. In the Supplementary Methods, section *Chromatin Immunoprecipitation*, we specify that “4 to 5 libraries were pooled and loaded into a total of 13 lanes. 50 bp single-end reads (50SE) were sequenced in an Illumina HiSeq2500 machine, using v4 chemistry”. The reviewer is right in pointing that this information was missing for ATAC-seq and RNA-seq data. We have now included this information in the corresponding sections in the Supplementary Methods.

For ATAC-seq data, it now reads: “All 9 libraries were pooled and loaded in a total of 3 lanes. 50 bp paired-end reads (50PE) were sequenced on a HiSeq 2500 platform (Illumina), using v4 chemistry”. For RNA-seq data, it now reads: “Three technical replicates (R1, R2 and R3) were sequenced per sample. All 9 libraries per technical replicate were pooled and loaded into one lane. 101 bp paired-end reads (101PE) were sequenced in a HiSeq4000 machine” .

Regarding how we control for unwanted technical and biological variation, we discussed this aspect in the Supplementary Methods section *Normalization of gene expression and enrichment signals across species*. In brief, we used the ComBat function from the R package sva (version 3.22.0) to remove the technical variation associated with the inclusion of previously published data for the human cell line GM12878 (H1). Second, we developed an inter-sample calibration method which is based on the identification of internal reference controls (IRCs). The rationale behind this approach is similar to other methods which use control genes (such as housekeeping genes) to normalize expression levels between samples. In our case, we wanted to apply the same approach to normalize RNA-seq, ChIP-seq and ATAC-seq data, so we defined these IRCs as either genes or genomic regions whose expression or enrichment signal is constant across samples and from which a normalization factor can be derived. Supplementary Fig. 37 offers a schematic representation of the approach we employed. Supplementary Fig. 39 shows the performance of the inter-sample normalization methods. To assess the goodness of the method we measured the angle to the identity line ($x=y$). If the normalization procedure removed noisy variance effectively, differences between samples were expected to be reduced. This would reflect in regression lines closer to the identity line (Supplementary Fig. 38). As shown in Supplementary Fig. 39 panel a) and c), the ComBat function effectively removes the technical noise associated with human sample H1. Then, the normalization factors derived from the IRCs, correct for biological variation across samples. Note how, after the correction step, differences between biological replicates are smaller whereas it becomes more apparent the differences between sample O1 and the other samples, as expected given the particularities observed for this cell line which was largely hypomethylated (Supplementary Fig. 43). We further compared our method to the commonly used quantile normalization, and observed that our method handled better the greater inter-samples differences at extreme values (Supplementary Fig. 40 and 41).

Finally, principal component analysis showed that gene expression patterns using normalized expression levels recapitulate the known primate phylogeny (Supplementary Fig. 35). Likewise, principal component analysis using normalized histones and open chromatin signal at orthologous regulatory elements recapitulate the known phylogeny (Author response figure 2, below). As expected, the type of epigenetic signal is the major driver of variation followed by the species. PC1 clearly discriminates inactive (H3K27me3) from active histone modifications and accessible chromatin signals, PC2 separates activity-determinant signal (H3K27ac, H3K27me3 and H3K4me3) from elongating (H3K36me3) and H3K4me1 and ATAC-seq signals. PC3 separates H3K4me3 from the rest. Lastly,

PC4 reflects the evolutionary distances between species. Note how macaque samples (inverse triangles) locate on top of the bottom-right panel in Author response figure 2, followed by orangutan samples (triangles), gorilla samples (rhombuses), and chimpanzee (squares) and human samples (circles). This pattern is coherently reversed in H3K27me3 (grey) (note that here we are evaluating H3K27me3 signal in the context of regulatory elements, where H3K27me3 imposes a poised state, contrary to the influence of all other epigenetic marks). We can also see that the distinct evolutionary rates of the epigenetic marks from the different spreads throughout the y-axis. H3K4me1 (blue) is the most dispersed signal; H3K4me1 is traditionally considered a hallmark of enhancers which have been shown to evolve faster than promoters. H3K4me3 (red) and H3K27me3 (grey), are less spread. This goes in line with slower promoter evolution and points towards high conservation of poised states. H3K36me3 is the least spread, in concordance with strong evolutionary conservation of gene expression patterns. Altogether these observations recapitulate what is known about the species phylogeny and the relationships between epigenetic marks and corroborate the goodness of our data.

Author response figure 2. Principal component analysis of normalized histone modifications and open chromatin signal at orthologous regulatory elements recapitulate the known phylogeny. Colors represent histone modifications. Shapes represent different species. The proportion of variance explained is indicated in parenthesis.

6) The approach used to define weak/strong/poised CREs should be made clear in the main text and more clearly explained in the figures. They should explain clearly and more in detail how a weak enhancer is different from a poised one, and so on.

We thank the reviewer for their comment and agree that given the relevance of weak enhancers in our manuscript, it is worth including a more detailed explanation in the main text. We have now included a

more explicit description of the epigenetic signal associated with each type of regulatory element in the revised manuscript that now reads as follows (lines 107-113):

“At large, promoters have a positive H3K4me3/H3K4me1 ratio, whereas H3K4me1 is more abundant in enhancers than promoters. Regulatory elements with strong activities have an H3K27ac enrichment level similar to that of H3K4me3 in promoters and H3K4me1 in enhancers. Poised regulatory elements have a characteristic enrichment of H3K27me3. Finally, weak promoters and enhancers are associated with lower intensity enrichments of all epigenetic signals and generally lack H3K27me3 and H3K36me3 (Supplementary Fig. 3).”

We have also expanded Fig. 2 caption to better explain the differences between each class of regulatory element. The description of Fig. 2a now read as follows (lines 969-978):

“Approach followed to annotate and classify regulatory elements (RE). In short, we classify RE in promoter and enhancer states with three activity levels (strong, poised or weak) based on a combination of chromatin marks and ATAC-seq signals. Bars represent relative enrichment of epigenetic signals in RE, from left to right: H3K4me3 (dark blue), H3K4me (light blue), H3K27ac (green), H3K36me3 (yellow), H3K27me3 (red) and open chromatin (grey). Promoters have a positive H3K4me3/H3K4me1 ratio, whereas H3K4me1 is more abundant in enhancers. Regulatory elements with strong activities are associated with high H3K27ac levels and poised regulatory elements have a robust enrichment in H3K27me3. RE are then linked to genes based on 1D gene proximity and 3D published chromatin maps for LCLs. RE not associated with any gene are referred to as orphan RE. See Methods and extended representation in Supplementary Fig. 1.”

Reviewer #3 (Remarks to the Author):

García-Pérez et al. have considered genome-wide data sets for the LCLs of 5 primate species (human, chimpanzee, gorilla, orangutan, and macaque), namely 5 histone marks (H3K4me1/3, H3K36me3, H3K27ac/me3), ATAC-seq, WGS, WGBS and RNA-seq. These data sets are extensive, and will be of value to others. The diverse analyses appear to have been performed to a high level. The study's results are consistent with what exists in the literature for more divergent species sets (e.g. repurposing of regulatory elements ref 18; Enhancer evolution across mammals ref 7; gene expression evolution ref 9). The two aspects that most caught-my-eye were: (a) that the 15 variable GLM explained two-thirds of gene expression variation, and (b) the discussion about positive selection of DNA variants in regulatory elements.

We thank the reviewer for appreciating the work and value of this project.

Major comments.

(1) The analysis of positive selection is the weakest aspect of this study. It appears that the genes that are being considered as being under “positive selection” are from refs 27-29. These publications discuss “accelerated evolution” (rather than positive selection) whose rationale could be GC-biased gene conversion (a phenomenon associated with recombination) and other mutational biases. Indeed, are the three human-specific intragenic enhancers that accumulate more hSNCs than expected altered by GC-biased gene conversion? If so, then the substituting base would be biased towards G or C (Glemin et al. *Genome Res* 25(8): 1215).

Furthermore, rapid evolution can be caused by relaxed purifying selection and this needs to be acknowledged where relevant, perhaps including where “the hSNCs density is higher in human-specific intragenic enhancers”. Rather than using these gene lists which are susceptible to alternative conclusions (e.g. FOXP2, Atkinson et al. *Cell* 174, 1424), the authors should use a more up-to-date data set, for example the genome-wide S-score generated by Green et al. (*Science* (2010) 328, 710) that made use of the Neandertal genome sequence.

We thank the reviewer for commenting on this aspect of our analysis. As stated by the reviewer, three out of the four datasets we were using for this analysis consisted of regions with signals of human acceleration rather than positive selection. Following the reviewer suggestion, we have now updated our dataset, including regions with signals of selection provided by the S-score as well as three other, more recent sets of regions that were putatively under positive selection on the modern human lineage¹¹⁻

¹³. We want to emphasize that the dataset used previously in our analysis (Peyr gne et al., 2017) is the most recent genome-wide analysis of such pattern of selection and making use of archaic genomes as well as large resources of present-day human variation, hence the one we may put a rather higher confidence compared to the dataset by Green et al. 2010. Taken together, these datasets are likely the current state of knowledge on putative positive selection in modern humans. We caution that since we use great ape and macaque LCLs, the timeframe of regulatory evolution is likely up to 25 Mya, which may be represented by accelerated regions using great ape genomes, while the studies on positive selection focus on a more recent timeframe of human evolution since the split from Neandertals (up to 800kya), for which no LCLs are available. We now explicitly describe the nature of the regions included in our dataset as a composite of regions with either signals of positive selection or acceleration (lines 384-386), briefly “putatively selected or accelerated regions” and we also acknowledge the possibility that relaxed selection is causing the pattern we observe (lines 494-495).

The reviewer was also concerned that the regions we were considering for this analysis could be influenced by GC-biased gene conversion (gBGC). According to their respective source papers, all three datasets had already been controlled for gBGC. However, we realized this could still be a potential issue affecting our results. To account for this, from our final updated dataset of regions with signals of positive selection and acceleration, we filtered out all those regions that overlapped with gBGC tracts as predicted by phastBias in the human genome¹⁴. By doing this, we restricted all downstream analyses to consider only those regions not influenced by gBGC. After filtering, we went from 1,907 to 1,722 regions, this means less than 10% of the initial regions were affected by gBGC. We then associated these regions with 4,747 unique genes and repeated the enrichment analyses to find the same trend as the one reported in the previous version of the manuscript (Author response table 1). We show the updated numbers in Fig. 6a.

Author response table 1. Enrichment results after removing regions that overlap with gBGC tracks (One-tailed Fisher’s exact test).

Datasets compared	OR	CI (95%)	P-value
gE vs RE ¹	2.2	1.90-Inf	7.77e-35
wE gE vs gE	1.5	1.30-Inf	5.16e-07
hswEgE vs wEgE	2.1	1.50-Inf	5.51e-05
hswEgE vs RE ¹	3.5	2.5-Inf	3.33e-11

¹Most general background: considering any gene linked to any regulatory element.

Also, we checked that none of the three human-specific intragenic enhancers with weak activities (hswEgE) and with a higher density of human single nucleotide changes (hSNC) than expected overlapped with the phastBias tracks either. Then, and following the reviewer’s suggestion, we asked

ourselves whether the high density in hSNC that we observed in hswEgE was also influenced by gBGC. There was not an enrichment of strong (S: G and C) over weak (W: A and T) bases in hSNC from hswEgE compared to all other hSNCs. However, this would not necessarily mean there is no gBGC in these regions, since an excess of W to S mutations in a predominant AT context could have given the same results. To account for this, we considered the ancestral and derived nucleotides in each substitution. We evaluated the enrichment of the W to S mutations compared to the complementary mutations (S to W mutations) considering the ancestral nucleotide state the one found in all other non-human primates included in our study and the derived state the one found in humans. We found no enrichment of W to S mutations in hswEgE compared to all remaining hSNCs (Fisher's exact test: $P > 0.05$, $OR = 0.9$). We have added these results in the manuscript (lines 402-404).

Taken together, these new analyses indicate that hSNCs in hswEgE are not particularly influenced by gBGC and thus confirm the findings we report in the manuscript.

(2) The authors should take extra care not to over-interpret their epigenetic data. For example, they argue that “These results confirm that the epigenetic activities of genic promoters, intragenic enhancers, and their interactions are likely the most direct determinants of gene expression regulation in our regulatory architectures.” Nevertheless, epigenetic marks are known, in some contexts, to be downstream consequences (and not upstream causes) of genic or enhancer activities so this statement needs toning down. They should be more explicit that the elements that are annotated are only putative regulatory elements, because some will not contribute to cellular or organismal phenotypes. Similarly, they should not imply causation when only correlation is known (e.g. “The number of active enhancers located near a gene – its regulatory complexity – has also been reported to influence the conservation of gene expression in mammals”). Similarly, the partial correlation analysis (e.g. Figure 4e) is not necessarily reflective of causation and this needs to be stated explicitly.

We thank the reviewer for these valuable comments.

The reviewer is right, the presence of specific histone marks is more precisely interpretable as proxies of regulatory activities with specific cases reflecting different relationships with gene regulation. The role in gene regulation of promoters and enhancers associated with the presence of these histone marks has already been shown^{15,16}. In existing literature, the regions detected with these histone marks are either considered as *putative regulatory regions* or as *regulatory regions detected with a given degree of uncertainty* (established in previous studies). Using one or other nomenclature often depends on the scope and resolution of the study. Following the reviewer's suggestion, we have now clarified this point both in the main text and Methods section and acknowledge the uncertainty of the regulatory elements we identify by referring to them as “putative” elements. For example:

Lines 92-94: “We used the signal of the ChIP-seq experiments from the five histone marks to identify putative regulatory regions with characteristic marks of promoters or enhancers (Supplementary Figs. 1 and 2).”

Lines 520-521: “Definition of regulatory elements: This analysis was performed to identify putative regulatory elements in each primate species.”

Similarly, our partial correlation analysis (as previous simpler covariation observations) does not prove causality, but it does show which regulatory elements present the stronger direct statistical co-dependence with gene expression. This direct co-dependence is compatible with causal relationships but does not provide information about the direction of the possible causality. In our case, we explain our results in the context of the epigenetic signals associated with each type of gene component. As mentioned before, we are aware our results do not prove causality, but combined with previous knowledge, they might suggest its presence.

However, the reviewer’s comment encouraged us to tone down the main text. We have rewritten the sentences pointed out by the reviewer (and other similar statements). Examples:

Lines 281-284: “These results suggest the epigenetic activities of putatively strong and poised genic promoters and intragenic enhancers and their interactions likely have a large influence on gene expression regulation in our regulatory architectures.”

Lines 285-289: “Although we cannot infer causality from our SPCA analysis, these networks reflect that regulatory co-dependencies between components depend on the distance of the elements in the network of chromatin contacts (with genic promoters and intragenic enhancers being in the gene locus, promoter-interacting enhancers interacting directly, and enhancer-interacting enhancers interacting indirectly with it).”

Lines 467-469: “In brief, coordinated epigenetic activities in genic promoters and intragenic enhancers form the core of these architectures and are strongly correlated with gene expression levels.”

(3) I assume that the authors will make the data sets readily available. I could not find a statement in the submission to this effect. Are all annotations being made available, e.g. PiEs and EiEs?

We thank the reviewer for raising this very important point. Indeed, all data has been made accessible in the SRA with the accession number PRJNA563344. We have now included a *Data availability* section at the end of the methods that reads (lines 925-928):

“Data availability

The raw fastq files from the genomic, transcriptomic and epigenomic data generated and used for the analyses in this study were uploaded to the Sequence Read Archive (SRA) with the BioProject accession number PRJNA563344.”

Additionally, all information regarding the putative regulatory elements we identify in each species, including both their epigenetic and type of component classification, is provided in Supplementary Table 1.

Minor comments.

We truly appreciate the reviewer’s effort of carefully reading the manuscript, going through our extensive supplementary sections and pointing several typos and minor comments. We have corrected these in the revised manuscript.

(4) Figure 2. Define aP and aE (“ambiguous”?) in the main text /legend.

We have added this definition in the legend of Fig. 2 (lines 979-981). Now it reads: “*aP and aE refer to ambiguous promoters and enhancers, respectively. Ambiguous RE are defined as those with a consistent state but different activities between replicates.*”

(5) “61% are 1-to-1 orthologous protein-coding genes in all primate species (Fig. 2c)” – explain that the 61% refers to REs and not genes.

We have clarified this sentence as suggested by the reviewer.

Lines 120-123: “*On average, 70% of the regulatory elements are associated with genes, of which an average across species of 93% of the regulatory elements are associated with protein-coding genes and 61% of the regulatory elements are associated with 1-to-1 orthologous protein-coding genes in all primate species (Fig. 2c).*”

(6) Typos: Suppl Figure 2 “Probabilites”;

We have corrected this typo to Probabilities.

Suppl Figure 8 “Number of regulatory elements with same the”. Is it simply by chance that 4 of 5 of the Wilcoxon p-values are identical?

Following one of the reviewer #2 suggestions, we have downsized our Supplementary Figures and only kept those figures we considered of most relevance. As a result, the figure this reviewer refers to is no longer included, although we have kept the analysis in the caption of Supplementary Fig. 5. However, Author response table 2 contains the DSC values associated with each type of regulatory element per species which were used in that analysis. It is by chance that the P-values obtained in the Wilcoxon signed-rank tests are the same.

Author response table 2. DSC scores of regulatory elements with the same regulatory state in both biological replicates of each species.

	Human		Chimpanzee		Gorilla		Orangutan		Macaque	
	pre-LDA	post-LDA	pre-LDA	post-LDA	pre-LDA	post-LDA	pre-LDA	post-LDA	pre-LDA	post-LDA
sP	0.85	0.9	0.85	0.91	0.73	0.83	0.82	0.87	0.88	0.93
pP	0.43	0.6	0.49	0.62	0.34	0.48	0.35	0.43	0.45	0.59
wP	0.3	0.32	0.38	0.51	0.31	0.44	0.41	0.49	0.35	0.41
sE	0.6	0.71	0.64	0.85	0.51	0.77	0.52	0.69	0.68	0.83
pE	0.43	0.49	0.65	0.68	0.5	0.51	0.53	0.5	0.71	0.76
wE	0.38	0.73	0.44	0.81	0.37	0.78	0.39	0.66	0.58	0.85
	$P = 0.015625$		$P = 0.015625$		$P = 0.015625$		$P = 0.03125$		$P = 0.015625$	

Suppl Figures 20, 21 and 22 – should the figure legends be identical?

We have collapsed previous figures 20, 21 and 22 in three different panels of what is now Supplementary Fig. 11. We have also revised the caption of the figure so the differences between the panels are clear. Now it reads:

“Supplementary Figure 11. Evolutionary dynamics of regulatory states. Left scatterplots show the patterns of evolutionary conservation for the different regulatory states (Kruskal-Wallis test, $P < 2.2 \times 10^{-16}$ in all). Right panels show the effect sizes of the pairwise comparison of the evolutionary conservation patterns between any two regulatory states (Dwass-Steel-Critchlow-Fligner test). Only the effect sizes of significant comparisons are shown ($P < 0.05$). The analyses are restricted to **a**, orthologous regulatory regions associated with genes; **b**, orthologous regulatory regions associated

with human protein-coding genes and c, orthologous regulatory regions associated with human non-coding genes.”

Suppl Figure 25 “phatsCons30 scores associated with TADs defined in cell the line”

We have revised what is now Supplementary Fig. 13 caption and corrected the typo.

Suppl Figure 30 “sates”.

We have corrected this typo in what is now Supplementary Fig. 15.

Explain in their legends how Suppl Figures 38 and 39 differ.

As part of our effort to make more manageable our Supplementary Material, and also following the advice of reviewer #2, we have removed most network Supplementary Figures, including what was Supplementary Fig. 39. These networks are still reported as Supplementary Tables 12-15. The difference between these networks is explained in Methods section *Partial correlation analysis*, more specifically in lines:

Lines 714-719: *“To better address this situation in our analysis, we defined a consensus signal for every component following the same approach established by WGCN³⁵ to define eigengenes as representative variables of clusters of co-expressed genes. In brief, we defined eigencomponents as the variables summarizing the common signals of the different histone marks in a component (actually calculated as the first PCA component of these five variables).”*

Lines 725-727: *“In addition, we defined the remaining unexplained signal of every histone mark by its eigencomponent as the residuals of a linear model of the original variables and the corresponding eigencomponent.”*

We have modified the legend of what is now Supplementary Fig. 22 and 23 to make more clear that these networks are constructed using the residuals from linear models of the original variables and the first eigencomponent of each regulatory component.

Suppl Figure 55 Wilcoxon signed rank test.

We have modified this in what is now Supplementary Fig. 25 and revised the naming consistency of all statistical tests used along the manuscript.

Suppl Figure 62, confirm that X-axis is hSNC density and Y-axis is number. Do not show smoothed data.

We have changed this plot in what is now Supplementary Fig. 32a to a cumulative frequency plot with X and Y axes properly labeled.

Figure 4d: is the X-axis a fraction, and not a percentage?

The reviewer is right about that. We are showing a proportion, so the X-axis now reads as “Proportion of RE”.

Line 337 “gene set shows”

We have corrected this typo in the revised manuscript.

(7) The authors should discuss whether there are any limitations in their approach of counting species versus using a phylogenetic approach.

As the reviewer points out, to evaluate the evolutionary conservation of regulatory elements we count the number of species in which any given regulatory state is conserved, regardless of the species phylogeny. We also explored human-centered evolutionary conservation taking into account the species phylogeny (Author response figure 3). We observe the same conservation trends, with slightly lower number of conserved elements in 2, 3 and 4 species, due to the exclusion of regulatory elements with possible incomplete lineage sorting (for example, conserved in human, gorilla and orangutan but not in chimpanzee), and lower elements at 1 species, since we only consider human-specific regulatory elements. In our manuscript, for the sake of clarity, we have favored the counting approach as a better representation of the observed global trends.

Author response figure 3. Evolutionary conservation of regulatory elements. a, Barplots show the average number of orthologous regulatory regions across species with the corresponding color-coded epigenetic state conserved in 1, 2, 3, 4 or 5 species. **b,** Barplots show the number of orthologous regulatory regions across species with the corresponding color-coded epigenetic state conserved in 1, 2, 3, 4 or 5 species according to the known species phylogeny.

(8) Noncoding gene models are likely to be most complete for human. Explain how this may change your study's results and alter its conclusions.

As noted by the reviewer, the non-coding annotation in non-human species is rather incomplete. This results in fewer regulatory elements associated with non-coding genes in the non-human species (Supplementary Fig. 18). The missing non-coding annotations in non-human species can create at least two different problems:

1. This unbalance could lead to regulatory regions associated with these genes in humans but remaining unassociated or even being wrongly associated with other nearby protein-coding genes in non-human species. To account for this, we considered the different quality of the assemblies and gene annotations to determine the genes associated with an orthologous regulatory region. Because of that, we assigned the gene associated with the orthologous regulatory element using the following hierarchy: human > chimpanzee > macaque > gorilla > orangutan. In this way, we minimize the effect of the poorest annotations in our associations. The very similar number of regulatory elements and activities associated with protein-coding genes in the different species, as well as the consistency of all our evolutionary and covariation analyses, show that this annotation unbalance does not significantly affect the regulatory networks of these genes.
2. More obviously, the absence of these non-coding genes in non-human genes makes it impossible to address the regulatory evolution of non-coding genes without an ortholog in humans. As this will bias any global analyses of non-coding genes, we restricted the analysis that involved non-coding genes to orthologous regulatory elements that could be associated with human non-coding genes:

Lines 164-166: “...*The regulatory architectures of protein-coding and non-coding genes –the latter evaluated in human due to underrepresentation of non-coding gene annotations in non-human species– show this same pattern...*”.

In consequence, we do not expect these limitations of non-human annotations to affect our results. However, it is clear that they underpower our ability to study the regulatory evolution of these genes in primates. In the future, when better annotations are available for these species, it will be exciting to investigate to what extent the evolutionary patterns observed in human non-coding genes are shared by other primates or if they are characteristic of humans. We foresee that our data and those provided by other studies will be fundamental to address this question when possible.

(9) Line 200-202 Remove: “Note also that conserved poised promoters are associated with very high conservation z-scores, which probably did not reach significance due to their low number (n = 9 pP)” because this is speculation.

We have now dropped this sentence from the revised manuscript.

(10) Figure 5. The authors need to better explain why, e.g., these findings in LCLs (an immune cell model) are related and biologically relevant to “developmental and proliferative functions” and

“neuronal functions”. The overarching conclusion (“These unexpected associations are likely to reflect the importance of particular epigenetic states in certain regulatory components to regulate specific processes”) was not particularly informative. I was not particularly convinced by this section.

We thank the reviewer for raising their concern about this section of the manuscript. As pointed out by the reviewer, it is indeed not expected to find functions related to development and brain in an LCL cell type. We are actually highlighting here that given that these are poised and weak states, which have a small impact in our cell type, they could be useful to study epigenetic conformations that are either conserved or human-specific and which, based on their functional enrichment, tissue-specific expression patterns and tissue-specificity scores of the associated genes, we hypothesize may have a more relevant functional impact in other cell types. The different combinations of conserved strong activities are, as expected, associated with house-keeping functions and/or with higher expression in LCLs compared to other cell types. This, in fact, indicates that our findings in the poised and weak states are equally consistent, though less intuitive: these regions could be associated with other functions and cell types. Moreover, as we complement the functional annotation analyses with the expression ones, we provide two independent analyses that point towards the same direction.

We agree with the reviewer that this section could be better argued and toned down. We have changed the title of this section to “*Different regulatory components with poised and weak enhancer states appear in LCL-unrelated genes and functions*”. In accordance, we have also changed the caption of Fig. 5 to “*Functional enrichment and gene expression patterns indicate that weak and poised enhancer states might be associated with LCL-unrelated cell types*” and clarified the rationale of our hypothesis in the discussion (lines 503-512). Finally, we have also modified the closing sentence the reviewer pointed out to the following (lines 370-373):

“These unexpected associations, along with the evolutionary conservation patterns of poised and weak enhancer states, suggest that these epigenetic states might be indicative of putative regulatory elements in other cell types different from LCLs.”

References

1. Guenther, M. G., Levine, S. S., Boyer, L. A., Jaenisch, R. & Young, R. A. A chromatin landmark and transcription initiation at most promoters in human cells. *Cell* **130**, 77–88 (2007).
2. Creyghton, M. P. *et al.* Histone H3K27ac separates active from poised enhancers and predicts developmental state. *Proc. Natl. Acad. Sci. U. S. A.* **107**, 21931–21936 (2010).
3. Rada-Iglesias, A. *et al.* A unique chromatin signature uncovers early developmental enhancers in humans. *Nature* **470**, 279–283 (2011).
4. Ernst, J. & Kellis, M. Chromatin-state discovery and genome annotation with ChromHMM. *Nat. Protoc.* **12**, 2478–2492 (2017).
5. Villar, D. *et al.* Enhancer evolution across 20 mammalian species. *Cell* **160**, 554–566 (2015).
6. Calo, E. & Wysocka, J. Modification of enhancer chromatin: what, how, and why? *Mol. Cell* **49**, 825–837 (2013).
7. Kim, T.-K. & Shiekhhattar, R. Architectural and Functional Commonalities between Enhancers and Promoters. *Cell* **162**, 948–959 (2015).
8. Ong, C.-T. & Corces, V. G. Enhancer function: new insights into the regulation of tissue-specific gene expression. *Nat. Rev. Genet.* **12**, 283–293 (2011).
9. Borsari, B. *et al.* Intronic enhancers regulate the expression of genes involved in tissue-specific functions and homeostasis. doi:10.1101/2020.08.21.260836.
10. Andersson, R. & Sandelin, A. Determinants of enhancer and promoter activities of regulatory elements. *Nat. Rev. Genet.* **21**, 71–87 (2020).
11. Racimo, F. Testing for Ancient Selection Using Cross-population Allele Frequency Differentiation. *Genetics* vol. 202 733–750 (2016).
12. Zhou, H. *et al.* A Chronological Atlas of Natural Selection in the Human Genome during the Past Half-million Years. *Cold Spring Harbor Laboratory* 018929 (2015) doi:10.1101/018929.
13. Prüfer, K. *et al.* The complete genome sequence of a Neanderthal from the Altai Mountains. *Nature* **505**, 43–49 (2014).
14. Capra, J. A., Hubisz, M. J., Kostka, D., Pollard, K. S. & Siepel, A. A model-based analysis of

GC-biased gene conversion in the human and chimpanzee genomes. *PLoS Genet.* **9**, e1003684 (2013).

15. Trizzino, M. *et al.* Transposable elements are the primary source of novelty in primate gene regulation. *Genome Res.* **27**, 1623–1633 (2017).
16. Zhou, X. *et al.* Epigenetic modifications are associated with inter-species gene expression variation in primates. *Genome Biol.* **15**, 547 (2014).

REVIEWERS' COMMENTS

Reviewer #1 (Remarks to the Author):

N/A

Reviewer #2 (Remarks to the Author):

The authors addressed all of my concerns.
The manuscript can be accepted.

Reviewer #3 (Remarks to the Author):

The authors have appropriately responded to all of my previous concerns.

A couple of typos:

line 206: "values (Dwass-Steel-Critchlow207 Fligner test)." (no extra full-stop)

line 314: "could have overrepresented combinations (Supplementary Fig. 26)"